# Predicting the effects of COVID-19 related interventions in urban settings by combining activity-based modelling, agent-based simulation, and mobile phone data

Sebastian A. Müller[1], Michael Balmer[2], William Charlton[1], Ricardo Ewert[1], Andreas Neumann[3], Christian Rakow[1], Tilmann Schlenther[1], Kai Nagel[1]*

1 Transport Systems Planning and Transport Telematics, TU Berlin, Berlin, Germany, 2 Senozon AG, Zürich, Switzerland, 3 Senozon GmbH, Berlin, Germany

* kai.nagel@tu-berlin.de

## Abstract

Epidemiological simulations as a method are used to better understand and predict the spreading of infectious diseases, for example of COVID-19. This paper presents an approach that combines a well-established approach from transportation modelling that uses person-centric data-driven human mobility modelling with a mechanistic infection model and a person-centric disease progression model. The model includes the consequences of different room sizes, air exchange rates, disease import, changed activity participation rates over time (coming from mobility data), masks, indoors vs. outdoors leisure activities, and of contact tracing. It is validated against the infection dynamics in Berlin (Germany). The model can be used to understand the contributions of different activity types to the infection dynamics over time. It predicts the effects of contact reductions, school closures/vacations, masks, or the effect of moving leisure activities from outdoors to indoors in fall, and is thus able to quantitatively predict the consequences of interventions. It is shown that these effects are best given as additive changes of the reproduction number R. The model also explains why contact reductions have decreasing marginal returns, i.e. the first 50% of contact reductions have considerably more effect than the second 50%. Our work shows that is is possible to build detailed epidemiological simulations from microscopic mobility models relatively quickly. They can be used to investigate mechanical aspects of the dynamics, such as the transmission from political decisions via human behavior to infections, consequences of different lockdown measures, or consequences of wearing masks in certain situations. The results can be used to inform political decisions.

**Data Availability Statement:** For computer code see https://github.com/matsim-org/matsim-episim. Simulations were computed with version

## Introduction

When COVID-19 took hold in Germany in February 2020, there was an urgent need for a differentiated modelling capability to predict the consequences of interventions. We used our

d16656f076640124de0361fc327d3803a80aa466 of the code, started with command java -jar matsim-episim-1.0-SNAPSHOT.jar runParallel –setup org.matsim.run.batch.BerlinSensitivityRuns –params org.matsim.run.batch. BerlinSensitivityRuns$Params. The input data (including the synthetic mobility traces) are made public here: https://doi.org/10.14279/depositonce-11495. The output data used for the figures can be retrieved at: http://dx.doi.org/10.14279/ depositonce-12113.

**Funding:** The work on the paper was funded by the Ministry of research and education (BMBF) Germany (01KX2022A) and TU Berlin. The BMBF Grant also funded the data provided by the commercial company senozon. MB and AN, employed by senozon, worked together with the rest of the team to iterate between input data and simulations until the input data contained all information needed to run the simulation. senozon provided support in the form of salaries for MB and AN, but did not have any additional role in the study design, data collection and analysis, decision to publish, or preparation of the manuscript. The specific roles of these authors are articulated in the 'author contributions' section.

**Competing interests:** MB and KN own shares of the commercial company senozon. MB and AN are employees of senozon. None of this alters our adherence to PLOS ONE policies on sharing data and materials.

experience with person-centric modelling of traffic [1] to build a first prototype within two weeks [2]. An advantage of using this starting point is that the whereabouts of all simulated persons, including their overlapping time spent at facilities or in (public transport) vehicles, are already given by the model, which is derived in part from mobile phone data. Since the input data contains age as an attribute of each synthetic person, it was straightforward to include agent-dependent disease progression into the model from the start. A short description of the different model variants over time is provided in S2 Text.

The model is regularly used to advise the German federal government (e.g. [3, 4]). The main contribution of those reports was and is to provide differentiated predictions of the influence of various interventions, such as reductions of activity participation, masks, or vaccinations. For the present paper, we show the contributions of different activity types to the infection dynamics as predicted by the model. We show how most activity types generate over time fairly constant contributions to the reproduction number $R$, independent from the actual *level* of R. In consequence it is structurally more stable to report reductions of $R$ caused by interventions as an additive term, rather than a term that is relative to the overall level of $R$ as is usually done (e.g. [5]). The model also explains why there are decreasing marginal returns to stay-at-home interventions [6]. Finally, the model makes a prediction concerning the magnitude of the difference between summer and winter, caused by moving activities indoors during winter.

## Related work

### Compartmental models

The general dynamics of virus spreading are captured by compartmental models, most famously the so-called SIR model, with $S$ = *susceptible*, $I$ = *infected/infectious*, and $R$ = *recovered* [7]. Every time a *susceptible* and an *infectious* person meet, there is a probability that the susceptible person becomes infected. Some time after the infection, the person typically recovers. Variants include, e.g., an *exposed* (but not yet *infectious*) compartment between $S$ and $I$.

Instead of running these models with compartments, one can run them on a graph [8, 9]. Persons are represented as vertices, connections between persons are denoted as edges. The random interactions that are implied by the compartmental models are then replaced by interactions with graph neighbors.

In reality, these interactions change from day to day; in particular, possible superspreading events like weddings or other large gatherings cannot be encoded in a static graph. For this, temporal networks have been investigated ([9], section VIII).

An advantage of compartmental models is that their runtime is independent from system size; in that way, it is easily possible to run a model for a country or a continent. A disadvantage is that one needs a separate compartment for each attribute combination (e.g. age × activity pattern × disease state), and that mechanical aspects such as the reduction of virus intake by masks, are difficult to include into the model. A special case is [10]: It treats each census block as a subpopulation, computes how virus travels from one census block to another via points-of-interest with visitors from both census blocks, and also has internal virus dynamics in each census block. The differences to our work are discussed in more detail in S5 Text.

### Person-centric epidemiological modelling

An alternative to compartmental models is to use synthetic persons as the starting point for modelling, and to "consider nodes as entities where multiple individuals or particles can be located and eventually wander by moving along the links connecting the nodes" [9]. Examples

of such models can be found since approximately 2004 [11–14]. A model of this type by Imperial College [15] had a large impact on policy in the UK. Other recent developments are [16, 17] on the global scale, or [18, 19] on the urban scale. These models typically follow individual synthetic persons. However, most of them, with the possible exception of the Virginia Biotechnology Institute model [12, 20], have explicit person movements only for commute patterns; all other infections are assumed to be in a local environment.

Aleta et al. [21] construct an agent-based model, similar to ours. Their data derives from persons specifically recruited to collect their long-term trajectories. They have long trajectories, with high spatial precision, but for only 2% of the population. This is still an impressive sample; however, with our work we aim for models where we have as many synthetic persons in the model as there are persons in reality.

A special case is by Kucharski et al. [22], who use a pre-existing dataset with recorded social contacts for 40 162 participants. This is close to our approach in that the persons who encounter each other for how long and in which context are microscopically specified. Differences include that it is not a model for the full population of a region, and the study does not trace behavioral changes throughout the pandemic.

## Daily activity trajectories

Using daily activity chains as the basis for transport modelling is an established approach in the transport modelling community. An activity chain is a sequence of activities of a person, where activities have types such as home, work, shop, etc., starting and ending times, and locations. There are several ways to generate such activity chains, for example by using activity-based demand generation models (e.g. [23, 24]), by taking them from travel diaries (e.g. [25, 26]), by using mobile phone data (e.g. [27]), or by data fusion from open access data sources (e.g. [28]).

In the present situation, we needed a technology that was readily available, allowed uniform rollout at least in Germany, and that would allow updates along with changes in mobility behavior during the unfolding of the COVID epidemics. For that reason, we used an established process that generates activity chains mostly from mobile phone data [27]. The process is described in more detail in S1 Text. The outcome of the process are activity chains, encoded as events (cf. Fig 1), for as many synthetic persons as Germany has inhabitants. Since the activity chains stem from transport modelling, they also contain knowledge about trips between activities, importantly trips by public transport, and in consequence also contain, for each synthetic person, events when they enter or leave certain public transit vehicles.

## Person-centric epidemiological models derived from transport simulations

From the section on person-centric epidemiological modelling above, one takes away that having person trajectories, and in particular where persons meet, would be useful for an epidemiological simulation. In consequence, the synthetic person trajectories from transport modelling explained above seem like a good starting point, since they are already available. Smieszek et al. [29, 30] and Hackl and Dubernet [31] construct epidemiological models on top of such pre-existing synthetic person trajectories; these are the main starting point for us. Najmi et al. [32] start from a person-centric transportation planning model for Sydney, and add a disease transmission model that computes possible infections based on co-locations during the simulated day. The approach is similar to ours, but does not use mobile phone data to track the actual mobility behavior. They also do not use an infection model that depends on the spatial situation of the activity type.

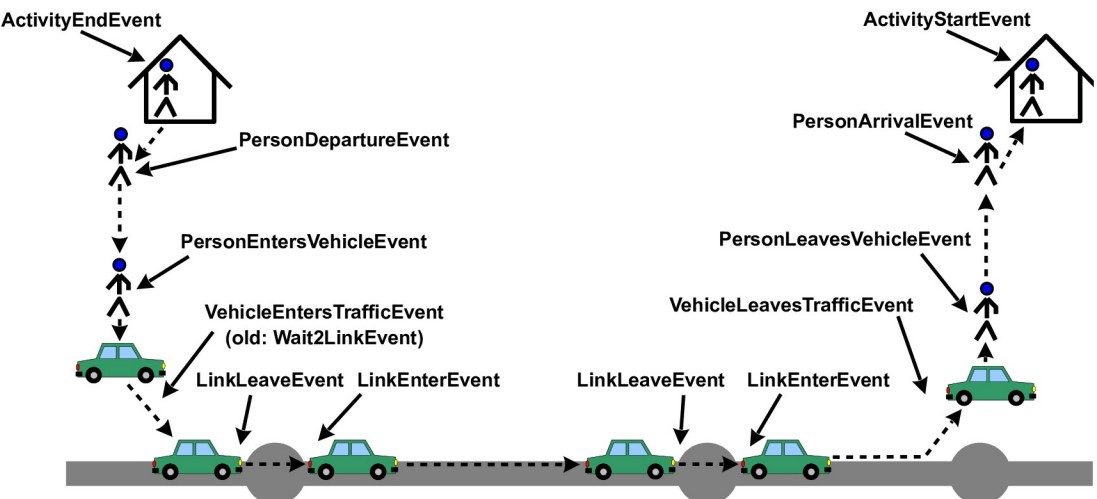

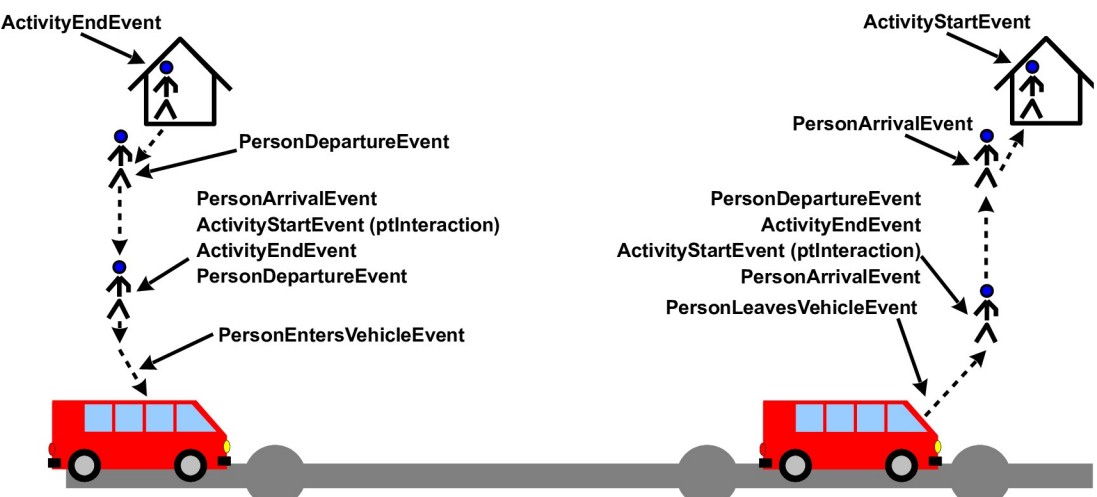

**Fig 1. Events for travel.** TOP: By individual vehicle. BOTTOM: By public transport. Source: [1].

The model described in the present paper has always been open source, and earlier version have been described in preprints [2, 33, 34]. This has been picked up by by Manout and Ciari [35] for Montreal, and by Bossert et al. [36] for South Africa. It was also used as the "micro" part by [37].

## From reductions of mobility behavior to reductions of infections

There are many data sets that track or analyze mobility changes "during Corona" [38–43]. This is, however, not our primary focus; rather, we are interested in how the infection dynamics can be better understood and possibly predicted with the help of mobility and other data. A possible approach to achieve this is data mining [44, 45]. We are, however, interested in models with more detail.

Jia et al. [46] and Xiong et al. [47] look at how long distance travel influences the disease import; they find that a high inflow from areas with high incidences is positively correlated

with high infection numbers. They do not, however, look at disease spread within the urban fabric, driven by daily movement patterns.

Fairly close to our work are Chang et al. [10], already mentioned earlier. They first construct, based on mobile phone data, a mobility network between census block groups and points of interest based on mobile phone data, and then use that model to investigate reopening strategies. They have a very detailed resolution of the facilities (they differentiate, e.g., between full-service restaurants, limited-service restaurants, and cafes/snack bars), but on the other hand they do not simulate individual synthetic persons. Similarities and differences are discussed in S5 Text.

## Model details

Important sub-models of agent-based epidemics models are: contact model, infection model, and disease progression model. These are described in more detail in the following sections.

### Mobility model and resulting contact model

As stated, we take the synthetic persons and their movements from transport modelling, cf. Fig 1. For the present study, the data is generated by a synthetic method developed by Senozon, see S1 Text. We have used and are using the same data for other projects [48–51]. From these activity chains, we extract how much time people spend with other people at activities or in (public transport) vehicles. That is, infection opportunities are directly taken from the input data. Details, for example of how multiple days or weekends are modelled, are provided in Sec 1 of S1 Appendix.

### Infection model

Once two persons are identified to have contact, and one of them is contagious and the other is susceptible, there is a probability of an infection. For this, we use the mechanical model by Smieszek [45]: infected persons generate a "viral load" that they exhale, cough or sneeze into the environment, and people close by are exposed. Overall, the probability for person $n$ to become infected by this process in a time step $t$ is described as

$$p(infect|contact)_{n,t} = 1 - \exp\left( -\Theta \sum_m sh_{m,t} \cdot ci_{nm,t} \cdot in_{n,t} \cdot \tau_{nm,t} \right) \qquad (1)$$

where $m$ goes over all other persons with which the person has contact at time $t$, $sh$ is the shedding rate ($\sim$ microbial load), $ci$ the contact intensity, $in$ the intake (reduced, e.g., by a mask), $\tau$ the duration of interaction between the two individuals, and $\Theta$ a calibration parameter. The model of Smieszek has the advantage that it was specifically developed with our transport simulation in mind, but there are many models of the same type (e.g. [52, 53]).

For small values of the exponent and just one contagious person in the room, one can approximate Eq (1) as

$$p(infect|contact) \approx \Theta \cdot sh \cdot ci \cdot in \cdot \tau \ . \qquad (2)$$

We do not use this approximation in our computer implementation, but it helps understanding the following arguments. Fig 2 gives some intuition about when that approximation holds; evidently, the effect of Eq (1) is to saturate when the infection probability becomes large.

All parameters can be given in arbitrary units as long as those units are always the same since the units are absorbed by $\Theta$. If one wanted to use physical units, then one could

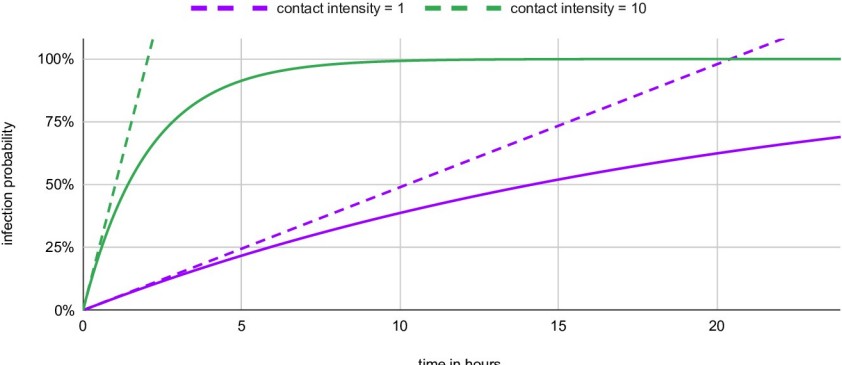

**Fig 2. How duration (in hours) translates into the infection probability for two different contact intensities.** The linear approximation of Eq (2) is given for either curve by a dashed line.

decompose $sh = \tilde{s}h \cdot out$ with $\tilde{s}h$ the, say, number of shedded virus particles per time. $out$ (for a mask on the shedding side), $ci$, and $in$ would be correction factors, i.e. 1 for a standard situation. $\tau$ would denote the time duration, so that the result would be the number $V$ of virus particles that were inhaled during that time duration. $\Theta$, or more precisely $1 - \exp(-\Theta V)$, would translate that number of virus particles into an infection probability. If that translation was known, one could attempt to calibrate the model from first principles. Practically, we use $\Theta$ as our main calibration parameter.

**Contact intensities.** For SARS-CoV-2, it is plausible to assume that a large share of the virus material is shed as aerosol [54]. In consequence, the first relevant term to compute the viral concentration in the air is the shedding rate, $sh$.

For such aerosols, it is plausible to assume that they mix quickly into the room, leading to the same uniform concentration everywhere [55]. Evidently, that concentration is inversely proportional to room size: if the room is twice as large, the resulting concentration is half as large.

Next, air exchange plays a role [55]. One could, for example, assume that the windows are opened once per hour, and all of the air is replaced with outside air. This would correspond to an air exchange rate of 1/h. If one assumes a constant rate of virus emission, there would be a linear increase of concentration up to the opening of the window, after which the virus concentration in the air would quickly go towards zero. The *average* virus concentration over this process would be half as much as the maximum concentration just before window opening. In consequence, the resulting average concentration is inversely proportional to the air exchange rate: If the air is exchanged twice as often, the resulting average virus concentration is half as large. This also holds for continuous air exchange, e.g. by mechanical means.

All of the above together replaces Eq 2 by

$$p(infect|contact) \approx \Theta \cdot \frac{sh \cdot in}{rs \cdot ae} \cdot \tau \quad, \tag{3}$$

where $rs$ is the size of the room, and $ae$ is the air exchange rate. That is, it sets the contact intensity $ci$ from Eq (1) to

$$ci = \frac{1}{rs \cdot ae} \quad. \tag{4}$$

**Table 1. Normalized contact intensities $ci'$, relative to the contact intensity at home, $ci'_{home}$.**

| activity type | area per person $fs$ [$m^2$] | air exchange rate old bldg $ae_{old}$ [1/h] | air exchange rate new bldg $ae_{new}$ [1/h] | share old buildings | resulting $ci'/ci'_{home}$ |
|---|---|---|---|---|---|
| home [58, 59] | 22 | 0.5 | 0.5 | | 1 |
| schools and day care [60] | 2 | 0.5 | 0.5 | 100% | 11 |
| universities | 4 | 0.5 | 0.5 | 100% | 5.5 |
| public transport [61, 62] | 0.33 | 2.0 | 10.0 | 50% | 10 |
| leisure [63] | 1.25 | 0.5 | 10.0 | 50% | 9.24 |
| shop | 10 | 0.5 | 1.5 | 10% | 0.88 |
| work [64–66] | 10 | 0.5 | 1.5 | 50% | 1.47 |
| errands | 10 | 0.5 | 1.5 | 50% | 1.47 |
| business | 10 | 0.5 | 1.5 | 50% | 1.47 |

Both the floor area per person and the air exchange rate come from building manuals or similar standards; note the given references in the table. The share of old buildings/vehicles is an estimate. Universities are assumed to have twice as much space per student as schools. Shop, errands, and business are assumed to follow the same characteristics as work. The contact intensities are computed separately for old and new buildings, and then averaged according to the assumed share of old buildings.

Again, the physical units are absorbed into $\Theta$; note, however, that the air exchange rate $ae$ is defined as the frequency of exchanging the air of the full room, and not of, say, cubic meters.

Aspects such as loudness of speech or if persons perform a physical activity are not taken into account in the present model although they are known to play important roles [56, 57], and they could, to some extent, be attached to the activity types. It is planned to include them in a future version of the model.

**Estimation of room sizes.** As stated above, our data resolves down to the level of "facilities". These correspond roughly to buildings. In consequence, such a facility can be anything from a single family home to a large office building or a sports arena. Since these facilities are too large compared to typical rooms, we divide facilities into $N_{spaces}$ rooms. $N_{spaces}$ is set to 20; the argument for this number is given in Sec 1 of S1 Appendix.

Since our simulation tracks when persons are at facilities, we can, for each facility, obtain the maximum number of persons at that facility, $N_{max}^{personsAtFacility}$, over the day. In addition, one can obtain typical floor space per person, $fs$, from regulatory norms and other sources (see Table 1). This leads to

$$facilityFloorSpace = N_{max}^{personsAtFacility} \cdot fs \ . \tag{5}$$

Divided by $N^{spacesPerFacility}$, this leads for the room size to

$$rs = \frac{facilityFloorSpace}{N^{spacesPerFacility}} = \frac{N_{max}^{personsAtFacility}}{N^{spacesPerFacility}} \cdot fs =: roomCapacity \cdot fs \tag{6}$$

where $roomCapacity$ is the maximum number of persons that are in the room during the day (thus its "capacity"); note that $N^{spacesPerFacility} = 1$ for home activities (cf. Sec 1 of S1 Appendix).

**Air exchange rate and normalized contact intensities.** Inserting Eq (6) into (4) results in

$$ci = \frac{1}{roomCapacity \cdot fs \cdot ae} =: \frac{1}{roomCapacity} \cdot ci' \tag{7}$$

with the "normalized" contact intensity

$$ci' = \frac{1}{fs \cdot ae} \quad . \tag{8}$$

See Table 1 for values of $ci'$.

$ci'$ parameterizes the "closeness" of the interaction. This is, cf. Eq (7), divided by *roomCapacity*, which denotes the number of persons that fit into the room given typical usage. If we share a room with one infectious other person, then our probability to become infected is, all other things being equal, half as large if the room is twice as large. However, if the room is twice as large, then there will presumably also be twice as many persons in it, doubling our own risk, and thus in the average cancelling out the effect of the larger room size. This second effect is computed directly by our contact model (Sec. From reductions of mobility behavior to reductions of infections above), and thus does not have to be included into the conditional infection probability. This has the additional advantage that if a person is in large container outside its peak usage, the model will calculate a much reduced infection probability. Examples for this are public transport vehicles, premises for large events, or restaurants.

A side effect of this model is that the above division by $N^{spacesPerFacility}$ has no effect in first order. If $n$ persons at the facility are all in one room, and one contagious person is added, the expected number of newly infected persons is $n \cdot p$, where $p$ is the individual probability to become infected according to Eq (1) with Eq (7). If the $n$ persons are divided between $N^{spacesPerFacility}$, and one contagious person is added into one room, the expected number of newly infected persons is $(n/N^{spacesPerFacility}) \cdot p'$, where $p'$ in first order is $p \cdot N^{spacesPerFacility}$, because *roomCapacity* in Eq (7) is divided by $N^{spacesPerFacility}$. In consequence, in first order the expected number of newly infected persons in the divided facility is the same, $np$, as in the undivided facility.—Second-order corrections come from the fact that Eq (1) eventually saturates when infection probabilities become large—then the smaller room sizes reduce the number of infections.

**Masks.**    The effectiveness of different mask types is taken from from [67], i.e. cloth masks reduce shedding and intake to 0.6 and 0.5 of their original values, surgical masks to 0.3 and 0.3, and N95 (FFP2) masks to 0.15 and 0.025. For some discussion of these values see Masks.

**Children.**    Current research implies that the susceptibility and infectivity are reduced for children compared to adults. We model this by including the susceptibility and infectivity into Eq (1). For adults both parameters are set to one. For people below the age of twenty the infectivity is reduced to 0.85 and the susceptibility to 0.45 [68, 69]. Note that this does not mean that the infection probability for children is necessarily lower than for adults, because children are more likely to perform activities with a high contact intensity, as shown in Table 1.

## Disease progression model

The disease progression model is taken from the literature [70–75] (also see [76]). The model has states *exposed, infectious, showing symptoms, seriously sick* (= should be in hospital), *critical* (= needs intensive care), and *recovered*. The durations from one state to the next follow log-normal distributions; see Fig 3 (LEFT) for details. We use similar age-dependent transition probabilities as [15], shown in Fig 3 (RIGHT).

Infecting another person is possible during *infectious*, and while *showing symptoms*, but no longer than 4 days after becoming *infectious*. This models that persons are mostly infectious relatively early through the disease [71], while in later stages the infection may move to the lung [72], which makes it worse for the infected person, but seems to make it less infectious to other persons.

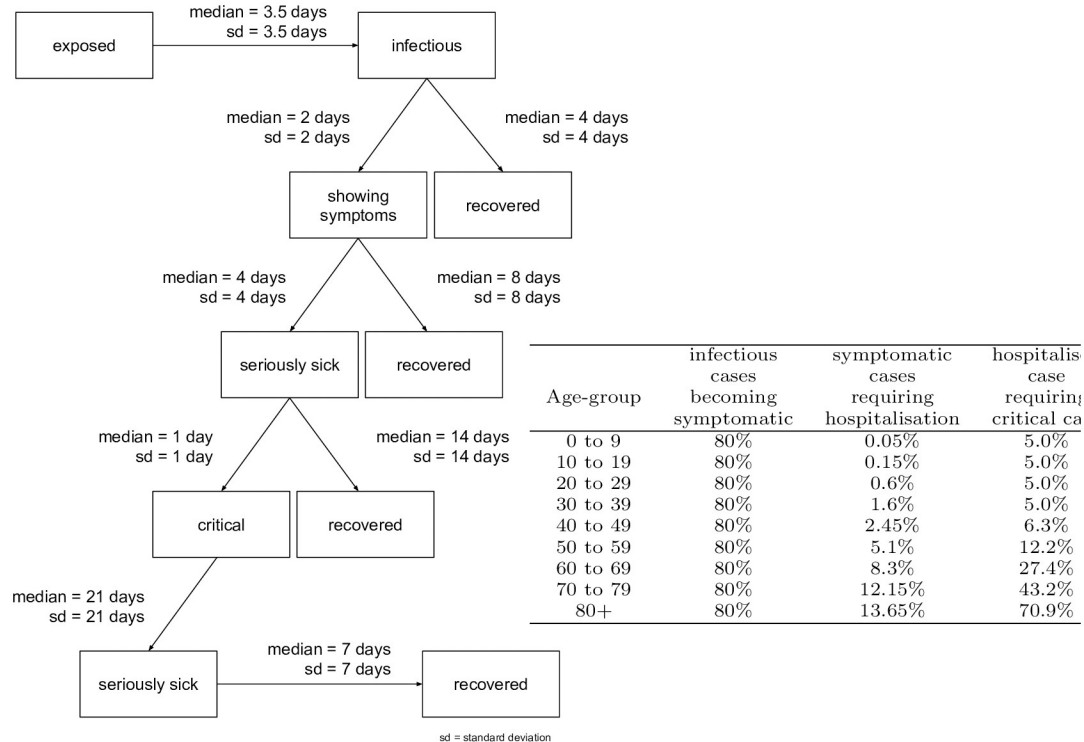

**Fig 3. Disease progression model.** LEFT: State transitions [70–75]. RIGHT: Age-dependent transition probabilities from infectious to symptomatic, from symptomatic to seriously sick (= requiring hospitalisation), and from seriously sick to critical (= requiring breathing support or intensive care). Source: [15], except that the numbers in the second column are divided by 2 (discussed in Under-reporting, and its variation over time).

## Time-dependent inputs and calibration

### Simulation runs

Although the approach was designed with uniform rollout throughout Germany in mind, the project, for reasons described in Sec 4 of S1 Appendix, mostly performed simulations for the metropolitan area of Berlin in Germany, with approx. 5 million people. A typical simulation run looks as follows:

1. One or more *exposed* (i.e. recently infected) persons are introduced into the population.

2. At some point, *exposed* persons become *infectious*. From then on, every time they spend time together with some other person in a vehicle or at some activity, Eq (1) is used to calculate the probability that the other person, if *susceptible*, can become infected (= *exposed*). If infection happens, the newly infected person will follow the same progression.

3. *Infectious* persons eventually move on to other disease states, as described in Fig 3.

The model runs for many days, until no more infections occur and all persons have finished their paths through the disease progression.

### Calibration

The calibration procedure undertaken for the present paper is described in the following sections. Calibration is performed by visual comparison, with first priority against the time series

of the number of hospital patients in Berlin, and with second priority against the COVID case numbers in Berlin. The calibration procedure, as described in the following, is as much about which elements to include at all as about finding the right parameters. "Second priority" here means that if calibration against hospital numbers is undecided between two alternatives, then the case numbers are used in addition. The case numbers are only used with second priority since the screening procedure has been changed multiple times, which means that the resulting time series is not homogeneous and thus not useful for model calibration. In particular, under-reporting in the initial phases was much larger than later. More information about the COVID case numbers in Berlin can be found in Sec 2 of S1 Appendix. A formal calibration of Θ can be found in Sec. Out of sample prediction. The calibration includes the following elements:

1. Calibration of the basic doubling time without reduction of activity participation

2. Integration of spring disease import

3. Calibration of the consequences of reduced activity participation

4. Calibration of an indoors/outdoors effect for leisure activities depending on the temperature

5. Integration of contact tracing, masks, and summer disease import

All calibrations concern Θ (cf. Eq 1); item 4 also involves defining threshold temperatures at which activities are moved outdoors at the end of the winter, and indoors at the end of the summer. All other aspects are data driven.

## Unrestricted model

Most parameters of the model are taken from the literature, as explained earlier, in particular Fig 3. The remaining free parameters are, from Eq (1), Θ, *sh*, and *in*. We have set the base values of $sh = in = 1$. As mentioned before, we use these parameters to model the wearing of masks, meaning that they are reduced when masks are worn.

Fig 4 shows the unrestricted base case with four different values of Θ. One finds that the aggregated behavior at this level corresponds to that of typical S(E)IR models, i.e. exponential growth, followed by a maximum, followed by exponential decrease. Based on these plots, theta-Factor values of 1.0 or 1.2 seem plausible to be consistent with the initial growth. A thetaFactor of 1.0 corresponds to Θ = 0.000561.

## Spring disease import

We take the disease import from abroad from data published by RKI ([77], always on Tuesdays). Currently, for Germany this data is only available on a nationwide aggregated level. For

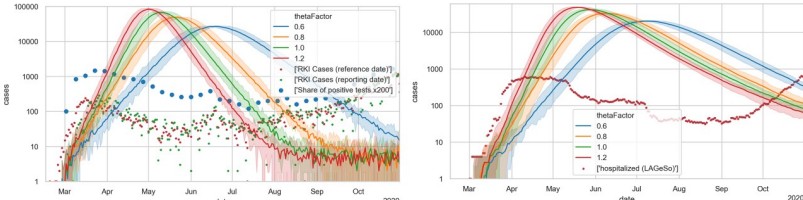

**Fig 4. Unrestricted base case.** LEFT: Case numbers. The green and red dots denote case numbers as reported by Robert Koch Institute [77]; the blue dots denote positive test fractions [78] multiplied by 200. RIGHT: Hospital numbers. Each simulation curve is averaged over 10 independent Monte Carlo runs with different random seeds; the shaded areas denote 5% and 95% percentiles of those 10 runs.

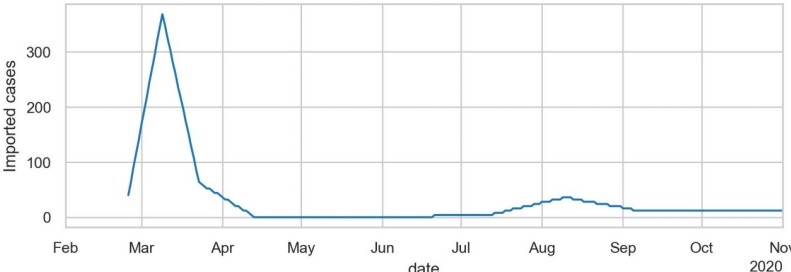

**Fig 5. Disease import over time.** Based on data taken from [77] (always on Tuesdays), but multiplied by 4 in spring, and divided by 2 in summer (see text for Discussion).

this reason we scale it down to our Berlin model by using the population size. The data is dated on the reporting date and not on the actual date of becoming sick. Since the infection seeds are initiated into our model with the status *exposed* (cf. Fig 3) and it can be assumed that the reporting date is significantly after the exposure date, we date the data from RKI back by one week. The data provided by RKI is available as weekly values so we assign these values to the respective Monday and then interpolate between them. Since we assume underreporting in the RKI numbers, we multiply them by 4; this is discussed in Sec. Under-reporting, and its variation over time. The initially infected persons are drawn randomly from the population. The resulting disease import is shown in Fig 5. The description so far only concerns the spring disease import; for summer disease import see Sec 3 of S1 Appendix.

An advantage about adding disease import is that the date of the first infection is no longer a free parameter: As shown in Fig 6, the disease import is sufficient to drive the first wave. The disease import data seems to lack some early cases, thus causing an initially nearly vertical increase in the simulation. The dynamics then settles onto the exponential increase shown in the previous section.

In terms of calibration, the initial growth is, within limits, insensitive against changes of $\Theta$, since it is dominated by the disease import. This can be explained by the fact that the exponential growth was running ahead in other areas, and in consequence the *share* of infected persons from those areas also grew exponentially. Only after travel was stopped did disease import also stop, and the dynamics in Berlin was dominated by internal processes.

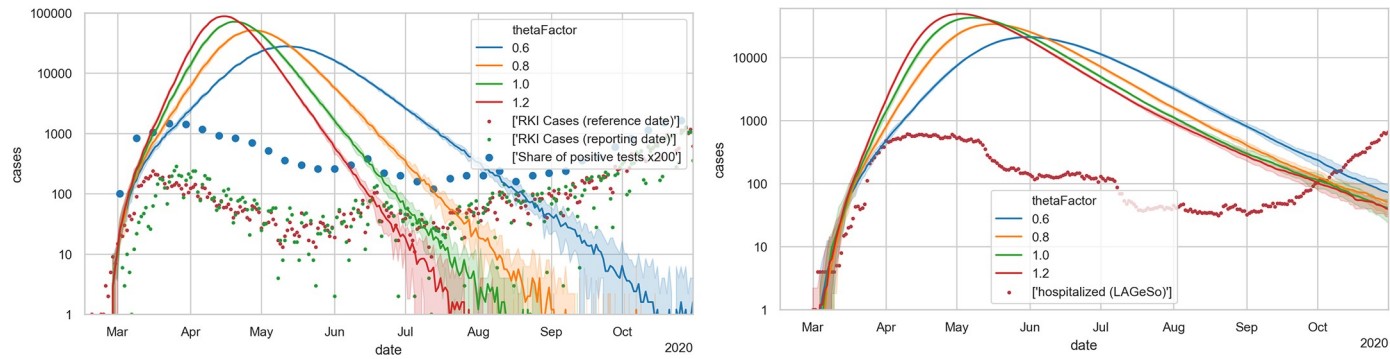

**Fig 6. Unrestricted base case, but with initial disease import from data.** LEFT: new cases; RIGHT: hospital occupancies. One finds that the initial slope dynamics is rather independent from the thetaFactor.

### Reductions of activity participation

During the unfolding of the epidemics, people decided or were ordered to no longer participate in certain activities. We model this by removing an activity from a person's schedule, plus the travel to and from the activity. For example, if a person in their original plan goes from home to activity A and then back home, then the activity plus both trips are removed from the schedule. If a person in their original plan goes from home via activity A to activity B and then back home, and activity A is deleted, then the following elements are removed: (a) the trip from home to activity A; (b) activity A; (c) the trip from activity A to activity B. In the current model, the schedule is not repaired: neither is the home activity nor are other activities prolonged, and also the trip chain is not mended. See S6 Text for possible improvements here. The consequence of those activity and trip removals is that the person no longer interacts with people at that activity location, and in consequence neither can infect other persons nor can become infected during that activity, or while in public transport vehicles to and from that activity. Overall, this reduces contact options, and thus reduces epidemic spread.

A very important consequence of our modelling approach is that we can take that reduction in activity participation from data. That data comes from the same source as our original activity patterns. However, the activity *type* detection algorithm is not very good for these unusual activity patterns during the pandemics, as one can see in S1 Fig when knowing that all educational institutions were closed in Berlin after Mar/15. What is reliable, though, is the differentiation between at-home and out-of-home time, as displayed in Fig 7. One clearly notices that out-of-home activities are somewhat reduced after Mar/8, and dramatically reduced soon

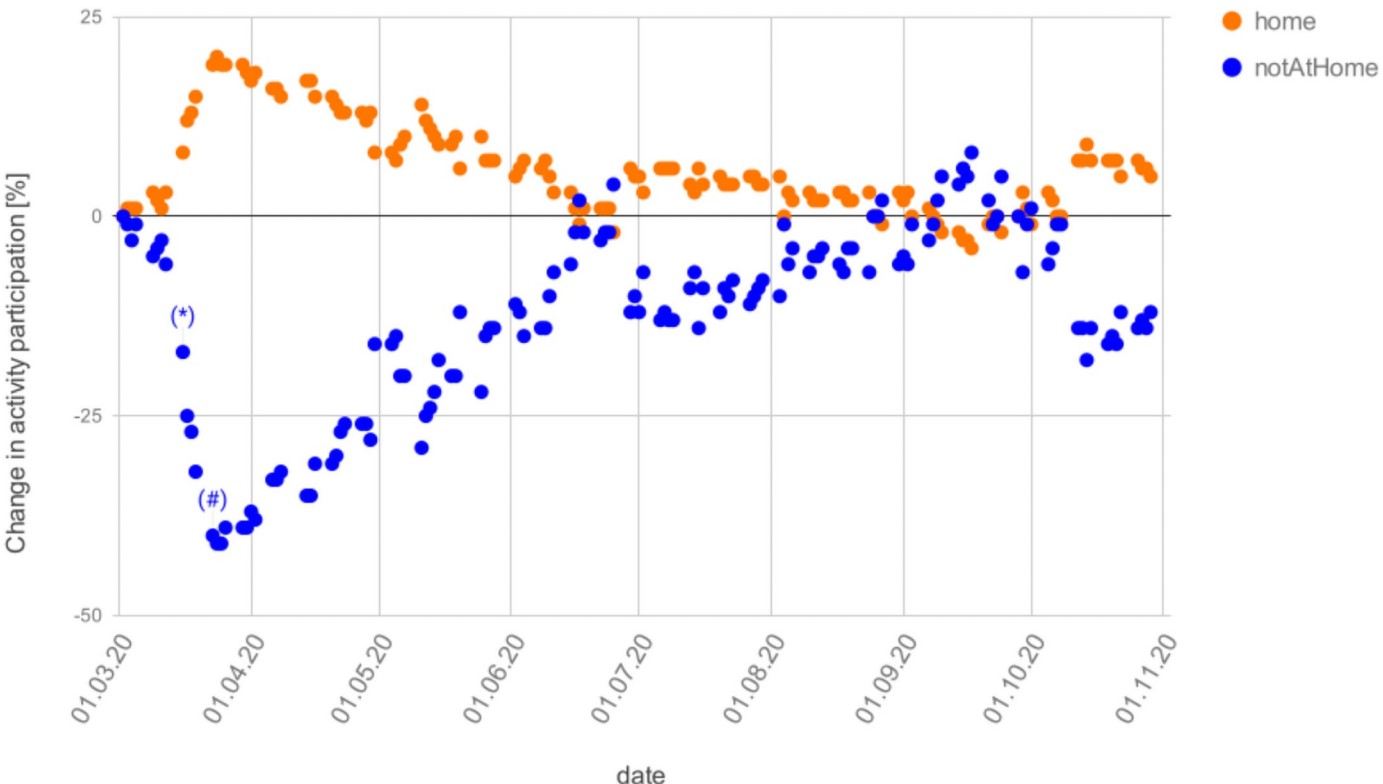

**Fig 7. Change in activity participation compared to the baseline for normal workdays.** All out-of-home activities are combined into one number. (*) denotes the first day of closures of schools, clubs, and bars; and (#) the first day of the so-called contact ban which came together with closures of all restaurants and non-essential stores.

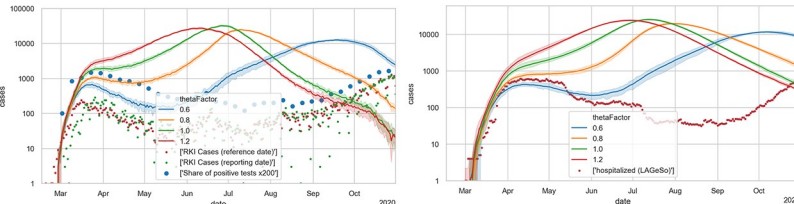

**Fig 8. Simulations with reductions of activity participation as obtained from mobility data.** LEFT: new cases; RIGHT: hospital occupancies.

after. After some experimentation, it was decided to take weekly averages of the activity non-participation, and use that uniformly across all activity types in our model, except for educational activities, which were taken as ordered by the government.

To remove an activity with a certain probability, a random draw is made every time a synthetic person has that activity type in its plan. This means that the model assumes that, say for a 50% work reduction, there will be a different 50% subset of persons at work every day. This intervention, in consequence, does not sever infection networks, but just slows down the dynamics.

One takes from Fig 8 that the mobility reductions, as given by the mobility data, is by itself not sufficient to explain the decreasing case numbers during spring. Evidently, one could now reduce Θ, and this is what we have done in our early simulations. This, however, artificially reduces the infection dynamics, and means that the simulation will miss the second wave in fall.

## Outdoors vs. indoors season

The probability of getting infected during an encounter depends on whether the encounter takes place indoors or outdoors. Outdoors, the probability of infection is significantly reduced compared to indoors. This is due to the fact that outdoors the air is constantly in motion and therefore aerosols cannot accumulate. We assume that an encounter outdoors decreases the infection probability by one magnitude [54, 79]. In countries like Germany, seasonality has a great influence on how much time people spend outdoors. In summer, people spend more time outdoors, while in winter they tend to spend more time indoors.

We include into our model that up to 100% of leisure activities are undertaken outdoors during summer, while that share reduces to 0% during winter. When an activity occurs outdoors, the otherwise identical computation of the infection probability is divided by 10. The model takes the actual temperatures as input; if the daily maximum temperature is larger than $T^* + 5C$, then all leisure activities that can happen outdoors are outdoors; if the daily maximum temperature is smaller than $T^* - 5C$, then all leisure activities happen indoors; in between, probabilities are linearly interpolated. We use $T^* = 17.5C$ in spring, linearly increasing to $T^* = 25C$ in fall; using a lower $T^*$ in spring is behaviorally plausible in Germany, and yields a far more plausible infection dynamics than keeping them the same.

The justification for this is as follows. A survey on physical activities [80] shows that, in summer, people in Germany perform about 80% of their physical activities outdoors, while this proportion shrinks to 10% in winter. We have assumed that other leisure activities (e.g. restaurants, visit friends) behave similarly. We also extend our range to 0 and 100% since the fluctuations of the temperature already lead to average values that are more than 0 and less than 100% (cf. Fig 9).

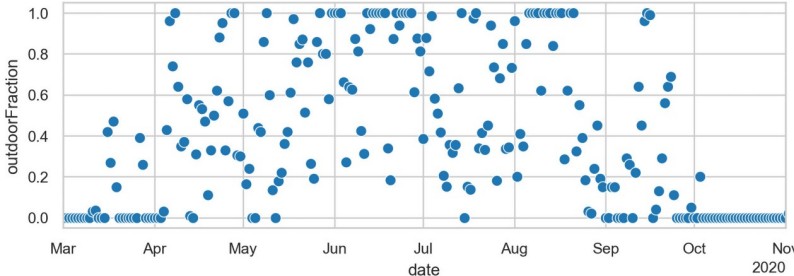

**Fig 9. Outdoors fraction for activities of type leisure, depending on the temperature of each day.**

Fig 10 shows an example of the infection dynamics where both $T^*$ in spring and $T^*$ in fall are 17.5C; as one can see, either the decrease of the first wave is not strong enough, or the second wave comes too late; note in particular the hospital numbers, which for all values of theta-Factor do not have enough slope in the second wave. The results with other $T^*$, as long as they are the same in spring and fall, are the same. Fig 11 shows instead using 17.5C for spring and 25C for fall; the second wave now is triggered earlier, and it is steeper. Fig 9 shows the outdoors fractions for this model.

There were some restrictions concerning leisure activities in place in fall. They mostly concerned large events. We know from our mobility data that all activities were at their normal level in September 2020; in consequence, if anything, they were divided into smaller groups.

## Masks, contact tracing, and summer disease import

From Fig 11 one takes away that a good calibration with the elements described so far would be possible, with a thetaFactor between 0.6 and 0.8. Nevertheless, we add masks (in public transport and shopping), contact tracing, and summer disease import, since they are plausible elements of the dynamics. In particular, they result in the prediction of reduced infection numbers for public transport and shopping, which both is plausible. This is described in more detail in Sec 3 of S1 Appendix.

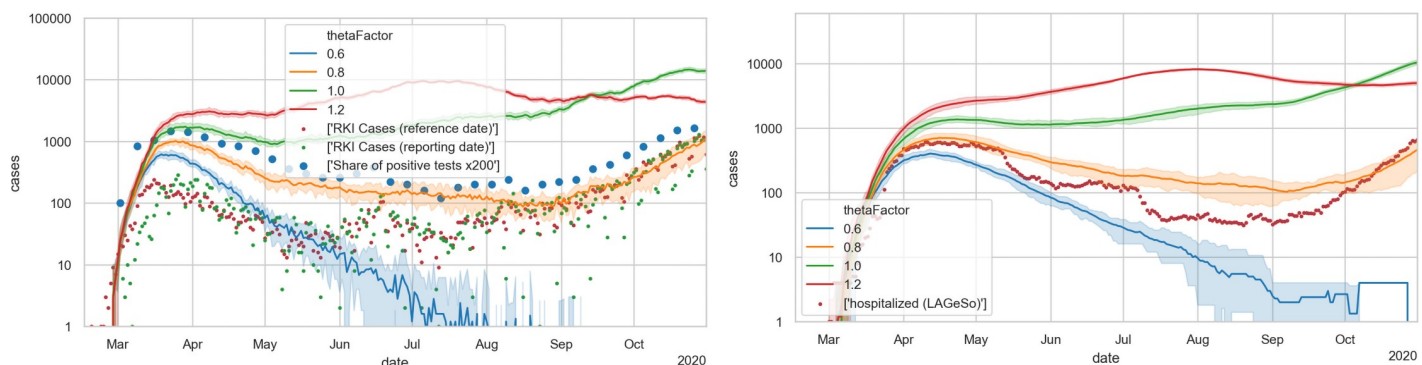

**Fig 10. Simulations that now also include a *symmetric* indoors/outdoors model, with a threshold temperature of 17.5C both in spring and in fall.** LEFT: new cases; RIGHT: hospital occupancies. A thetaFactor between 0.6 and 0.8 is most plausible, but the second wave would come too late (starting after September) and would not be steep enough (compare slope of red dots in right plot after September) (cf. in particular the hospital numbers).

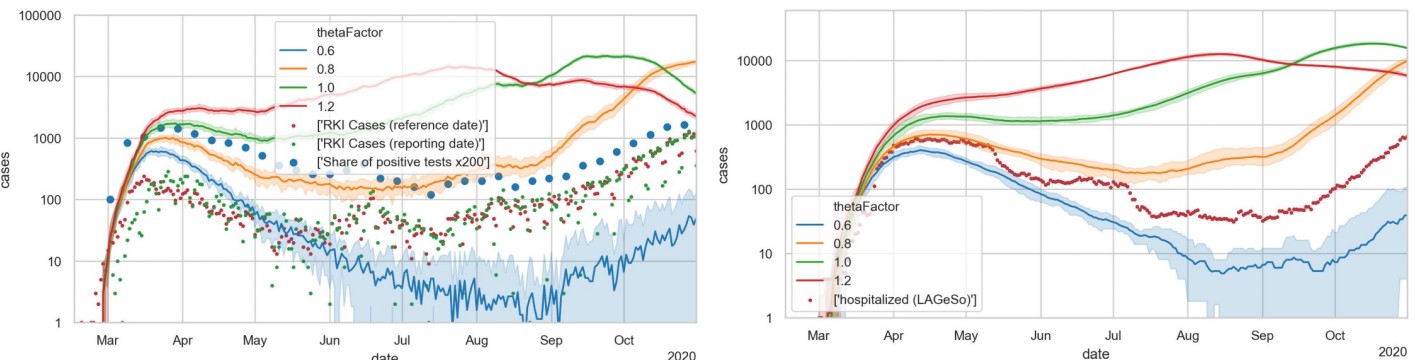

**Fig 11. Simulations that now also include an *asymmetric* indoors/outdoors model, with a threshold temperature of 17.5C in spring, and 25C in fall.** LEFT: new cases; RIGHT: hospital occupancies. A thetaFactor between 0.6 and 0.8 is most plausible, which would well reproduce the second wave (cf. in particular the hospital numbers).

## Final model

The final model is shown in Fig 12, where the blue line traces the number of new cases with state *showingSymptoms* from our simulation. Fig 12 (right) shows the cases in need of hospital care and those in need of ICU care from our simulation compared to real data. As stated, we find fitting to the hospital numbers more important; fully fitting to the case numbers is not possible with just one $\Theta$ that is constant across the whole simulation. Note that this implies, as stated, a strong deviation of the model curve (in blue) from the reported numbers (in red and green) during the first months. Also see Under-reporting, and its variation over time in the Discussion.

## Methods and results

### Infections per activity type

Evidently, in our microscopic models we can track how many infections happen at which activity type. Fig 13 shows, on top, the absolute numbers of infections per activity type for the simulation, and below the *share* of infections per activity type over time. To obtain these numbers, we evaluate what activity the infected person is performing at the time of infection and date that to the date of infection.

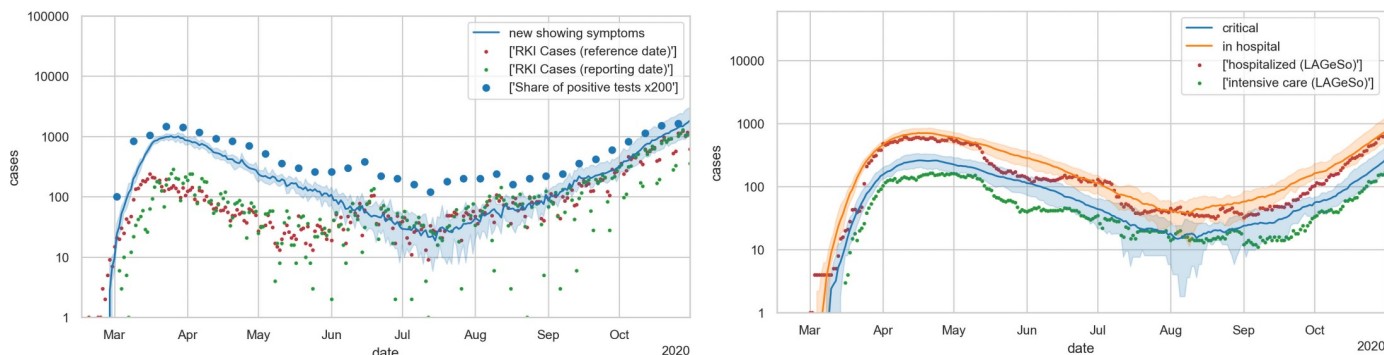

**Fig 12. Final model.** LEFT: new cases; RIGHT: hospital occupancies. All simulation results are averaged over 10 runs with different Monte Carlo seeds; the shaded areas denote 5% and 95% percentiles of those 10 runs. Evidently, the relative errors become larger with smaller case numbers. The simulation model can only be fitted against the hospital numbers (right) when significant under-reporting is assumed in the early phase (left).

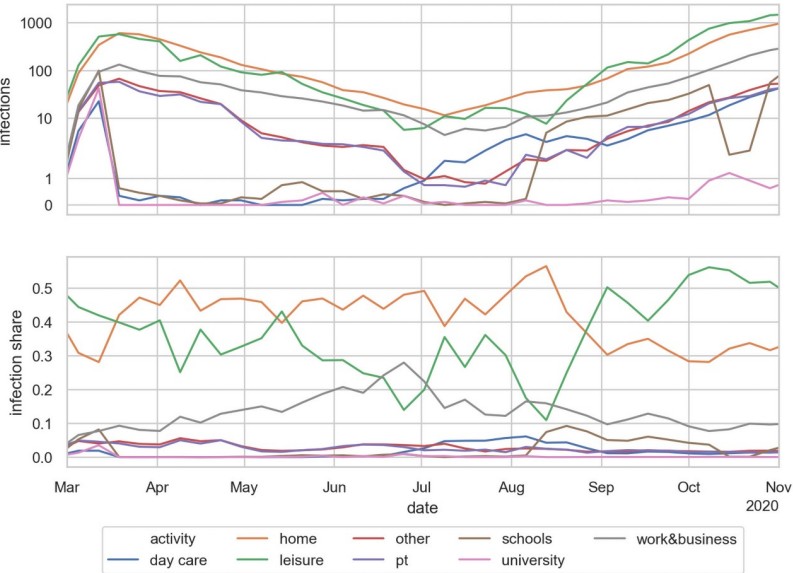

**Fig 13. Infections per activity type.** TOP: absolute numbers. Note logarithmic scale. BOTTOM: Share of infections per activity type. The values are averaged over the same 10 runs as for the other figures, and in addition aggregated into weekly bins. One can see, for example, the return to school near the beginning of August, and the fall vacations in October.

Initially, all activity types play a role. After the closure of the universities, schools, and day care in March, both their absolute numbers and their shares go to zero. At the same time, the infections share of work (gray) in April and May reflects that persons were drifting back to normal activity patterns (cf. Fig 7). Leisure (green) would have shown the same trend, but that was counter-acted by the increasing shift of activities to outdoors. In the bottom plot, the purple line shows how the share of infections in public transit decreases significantly near the end of April because of increased wearing of masks. (Recall that we use observed mask compliance.) In July we see how day care (blue) picks up, because it was re-opened. Schools re-open in the second week of August, and pick up accordingly (brown). Also, two weeks of school vacation in October are clearly reflected in the brown curve. From September on we then see a strong increase of the infections share of leisure activities—corresponding to moving leisure activities from outdoors to indoors as explained earlier.

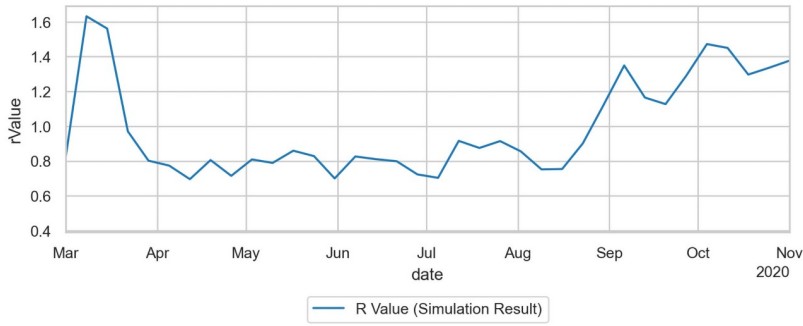

**Fig 14. Reproduction number $R(t)$ for the duration of the simulation.** As explained in the text, we explicitly count the reproduction number per agent, and then average them over all agents that turned contagious on a given day.

## Reproduction number

Since our method is person-centric, we can, for each infected person $n$, count the number of persons that that person infects, i.e. its reproduction number, $R_n$. When averaging over multiple persons, one needs to make a decision to which date $R_n$ is assigned. We use the date when $n$ turned contagious, and in consequence

$$R(t) = \sum_{n \in C(t)} R_n \ ,$$

where $C(t)$ refers to all persons who turned contagious on day $t$. An issue with this approach is that the consequences of interventions become visible in $R(t)$ before the interventions actually start—since the infections that are suppressed happen later than $t$. This is also the reason why we use the date when turning contagious and not the date when they got infected, since that would increase that temporal gap even more. Fig 14 shows the resulting values, with $R(t)$ much larger than one in the initial phase, then lower than one until the end of summer, and then increasing to above one in fall. We do not offer a comparison with the official $R$ values since they have the same issues as the official case numbers.

## Reproduction number per activity type

More insightful than the number or share of infections, as presented in Sec. Infections per activity type above, is the average *reproduction number* in each activity type. The method counts for each infected person the number of persons they infect at each activity context. As in Sec. Reproduction number above, the numbers are dated back to the date when the person became contagious, and then averaged over all those persons.

For example, an activity-specific R-value for school of $R_{school}(t) = 0.1$ means that each person that becomes contagious on day $t$, in the average, infects 0.1 other persons at school. Evidently, if only 10% of persons turning contagious on day $t$ have school anywhere in their activity pattern, then each such person would have to infect one other person in the school context in order to reach the population-average value of 0.1.

Adding up these activity-specific reproduction numbers leads to the overall reproduction number. This explains why, in first order, the overall reproduction number can be additively decomposed into the contributions of the different activity types.

One sees, in Fig 15, that the reproduction number at home remains roughly constant—a person who gets infected in any way reinfects on average about 0.35 persons at home. Work is related to the mobility data—if less time is spent out-of-home, then in the model less time is

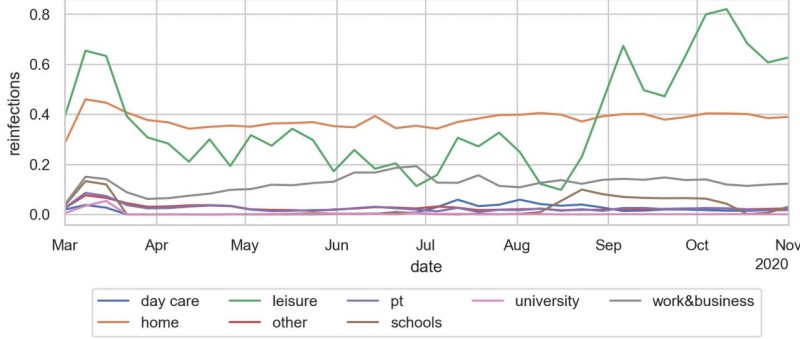

**Fig 15. Reproduction number per activity type.**

**Table 2. Contributions to *R* by activity type and intervention according to our model.**

|  | contribution to *R* |
|---|---|
| home | **0.44** |
| . . . with cloth / N95 masks | 0.20 / 0.02 |
| work | **0.17** |
| . . . @ 75% / 50% | 0.10 / 0.04 |
| . . . with cloth / N95 masks during work | 0.06 / 0.01 |
| . . . @ 50% with N95 masks during work | < 0.01 |
| schools | **0.15** |
| . . . @ 75% / 50% | 0.07 / 0.02 |
| . . . with cloth / N95 masks during classes | 0.05 / < 0.01 |
| . . . with N95 masks during classes and 50% attendance | < 0.01 |
| day care | **0.02** |
| . . . @ 75% / 50% | 0.01 / < 0.01 |
| . . . with cloth / N95 masks | 0.01 / < 0.01 |
| universities | **0.23** |
| . . . @ 75% / 50% | 0.11 / 0.03 |
| . . . with cloth / N95 masks | 0.06 / < 0.01 |
| retail and errands | **0.09** |
| . . . @ 75% / 50% | 0.06 / 0.03 |
| . . . with cloth / N95 masks | 0.03 / 0.01 |
| leisure (winter) | **1.04** |
| . . . @ 75% / 50% | 0.52 / 0.21 |
| . . . with cloth / N95 masks | 0.38 / 0.03 |
| leisure (summer) | 0.2 |
| public transport | **0.12** |
| . . . @ 75% / 50% | 0.06 / 0.03 |
| . . . with cloth / N95 masks | 0.04 / < 0.01 |

For these calculations we run the unrestricted model without any interventions and then introduced the interventions described in the left column on 2020–04-01. The reductions to the *R* values were calculated one week after that, comparing the respective weekly averages. For the mask interventions, the compliance rate is 90%.

spent at work, leading to fewer infections. Schools were closed in the middle of March, and not reopened until the second week of August. Also, there is a school vacation during the second and third week of October. Day care according to the model has little effect. Day care was already re-opened partially in June, and fully in July. The reproduction number at leisure is strongly driven by the weather: If it is warm, the model assumes that most of the leisure activities take place outdoors, where they contribute little to the infection dynamics. In consequence, this effect plays an important role in spring, where the warmer temperature played as much a role as the reduction of the out-of-home activites. One also clearly sees the strong growth of the leisure reproduction number in fall, which according to these simulations is driving the second wave in Berlin. Public transport is strongly visible in March, until the obligation to wear masks was introduced. All other infection contexts, e.g. errands or business activities, are combined in the category "other".

## Reductions of *R* per intervention

Other papers, e.g. [5], report, for various interventions, corresponding percent reductions of *R*. Our model clarifies that it is structurally more robust to report the *additive* reduction of the reproduction number by a certain intervention. For example, according to our model closing schools removes the school reproduction number from the dynamics, and in consequence reduces *R* by about 0.15. If *R* is 1 when the intervention is introduced, this amounts to 15%; if *R* is 2, then this amounts to 7.5%.

Table 2 shows, based on simulations as explained in the previous section, the contributions to *R* of the different activity types. Adding up the boldface numbers leads to *R* = 2.26, i.e. a strongly super-critical situation. In contrast, the 2020 Germany summer regime corresponds to closed universities, schools and day care, and wearing masks in retail. Together with the leisure summer number this leads to *R* = 0.88, i.e. makes the situation sub-critical.

It has been pointed out by other studies that the reproduction numbers at home play an important role and reduce the remaining "space" one has available for infections outside home [81]. The reproduction number at home can be reduced by moving persons showing symptoms, and more radically persons identified as contacts by contact tracing, into separate facilities, sometimes called quarantine hotels.

One also notices that all infection contexts can be strongly reduced by wearing masks—this (evidently) even holds for leisure. Clearly, they would need to be worn *during* the activities, and not just during access and egress. Wearing masks during class at school has hesitantly been adopted in Berlin during November; wearing masks during work, in particular in office buildings, has never been pursued seriously in Germany and is still not obligatory if occupants have at least 10 $m^2$ available per person—which is the value with which our simulations run and which generate the numbers of Table 2.

Evidently, a tricky context is leisure. According to our simulations, leisure alone, in conjunction with home, would be sufficient to keep *R* above one during winter, and thus needs to be suppressed accordingly. Keeping other activity contexts open without masks implies that leisure needs to be suppressed even further if *R* < 1 is to be achieved.

Conversely, during summer achieving an *R* < 1 is relatively easy. This explains why there were few problems during summer in Germany (and most other European countries). Evidently, all of this refers to the original variant of SARS-CoV-2, which was less transmissible than later variants.

## Decreasing marginal effect of interventions

In Table 2, for all activity types, a reduction of the participation by 50% reduces the contribution to R by far more than 50%: at work from 0.17 to 0.04, at school from 0.07 to 0.02, etc. In consequence, the next 50% reduction of participation, i.e. closing the activity type completely, will yield a much smaller reduction of infections. This is consistent with the empirical observation that the marginal effect of stay-at-home interventions decreases [6].

From our model, this can be explained as follows (see Fig 16): Assume, for example, that each morning each school child throws a coin and goes to school only when it shows heads; this means that school participation is reduced to 50%. In consequence, if there is an infectious person at school, only half as many other persons have a chance to get infected. (This assumes that they use the same classrooms as before, at half the density.) However, the probability that an undetected infectious person goes to school is also reduced to 50%. Multiplying these two probabilities means that only 50% · 50% = 25% of the infections happen in this case. That is, the first 50% of the reduction has already 75% of the possible effect.

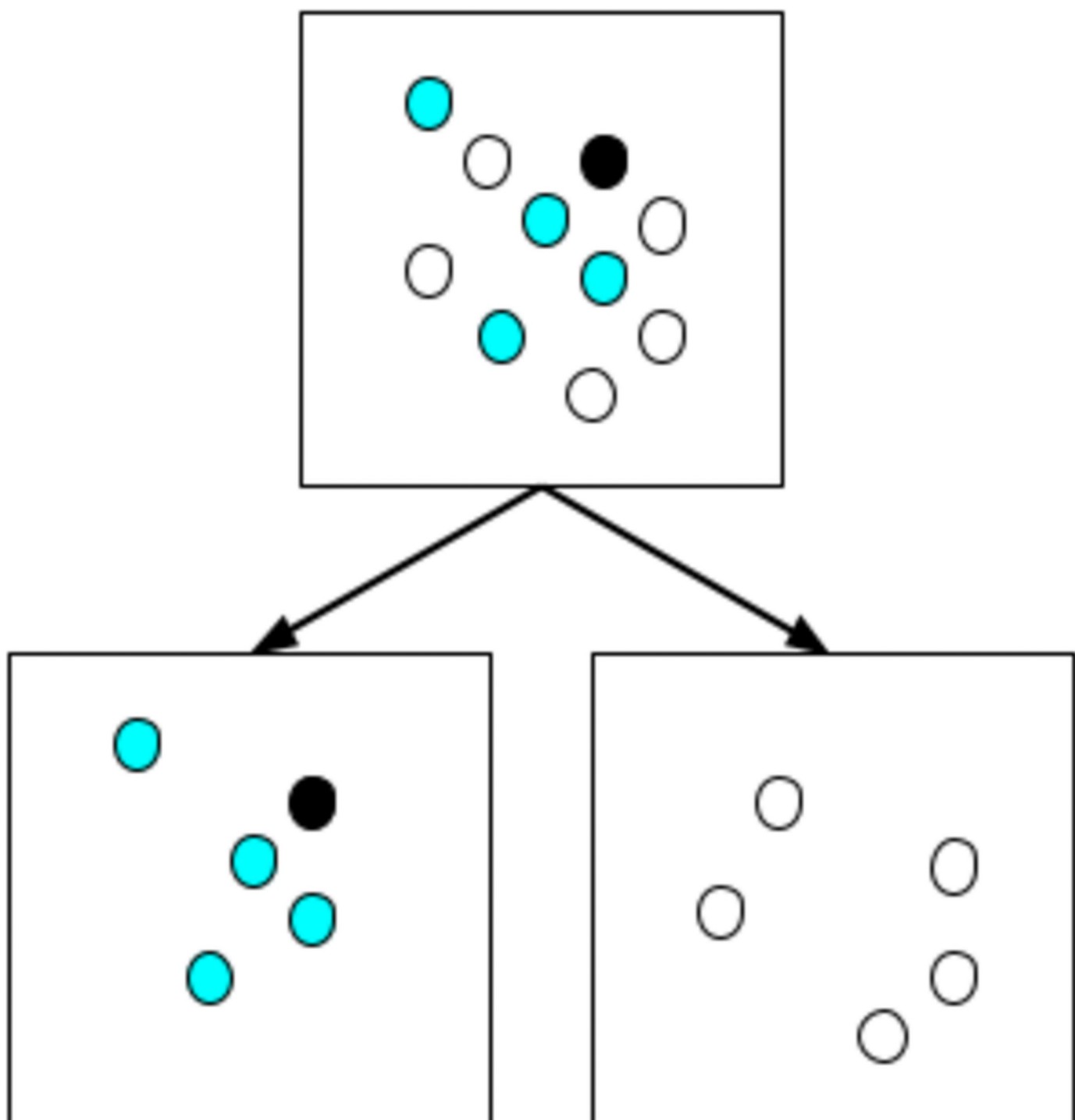

**Fig 16. Effect of dividing a group of 10 persons into two groups of 5 persons each.** In the original situation, each of the 9 susceptible persons (white and cyan) has a probability to get infected of $p_0$, resulting in a expected number of infected persons of $9p_0$. In the divided situation on the left, the expected number of infections is $4p_0$. On the right, it is 0. Overall, this results in an expectation value of $\frac{1}{2}(4\,p_0 + 0) = 2\,p_0$. In consequence, when dividing classes and alternating their attendance, the number of infections is reduced from $9p_0$ to $2p_0$. For large group sizes, the reduction converges to 1/4. The same holds when each individual attendance is decided randomly with probability 1/2 at the beginning of each day.

More generally, consider an activity in an enclosed space, with $N$ participants, $k$ of them contagious, $k \ll N$, and conditions such that the linear approximation of Eq (2) for the infection probability holds. In this situation, the expected number of infections is proportional to $kN$. Now assume that the participation probability at that activity, for each individual person, is reduced from 1 to $\alpha < 1$. There are two consequences:

1. The expected number of contagious persons reduces to $\alpha k$.

2. The expected number of participants reduces to $\alpha N$.

The expected number of infections in consequence reduces to $\alpha k \alpha N$, i.e. $\alpha^2$ as many as before.

Evidently, this means that $1 - 0.75^2 \approx 44\%$ of the effect is obtained with the first 25% of the intervention, another $1 - 0.5^2 - 44\% = 31\%$ of the effect are obtained with the next 25% of the intervention, and the remaining 25% of the effect need the remaining 50% of the stay-at-home intervention for this particular activity.

In terms of the management of COVID-19, this implies that it is far better to include each activity type/sector of the economy to some extent, rather than shutting down some sectors completely while leaving some other sectors completely open.

## Out of sample prediction

We show the predictive performance of our model by calibrating the simulation on a fixed training set and comparing simulation results into the future against unused data. In this calibration Θ is calibrated such that the Root Mean Squared Logarithmic Error (RMSLE) between hospital cases in the simulation compared to historic data is minimized. For this, the simulation is run with eight different Monte Carlo seeds and then the results are averaged. Because one simulation run is quite computationally expensive, a *Tree-structured Parzen Estimator* [82], implemented by the Optuna package [83] in Python, is used to sample the parameter space more efficiently.

RMSLE has the advantage that it is less sensitive to the scale of the data than RMSE. That is, relative errors in valleys have as much weight as relative errors on the ridges. This also corresponds to the visual impression of the logarithmic plots often used in epidemics and used throughout the paper. Results using RMSE instead of RMSLE and some more discussion can be found in S3 Text.

We run this calibration up to various dates. From there on, we perform two types of prediction: (a) Fig 17 left: With activity participation levels frozen at the level of the last calibration day (see the second column of Table 3; only during school vacations are school and work activities reduced for prediction dates); (b) Fig 17 right: With activity levels as given from the data also for prediction dates. For both cases, the import is frozen at 4 imported cases per day, while the daily maximum temperature is forecast based on the average over the last 10 years.

One finds that the correct activity level (Fig 17 right) is crucial especially for longer-term predictions: Even with calibration only to the end of April, the model predicts the autumn wave very well, while when the activity numbers are frozen (Fig 17 left), the forecasts have a drift depending on whether the activity level is too low or too high on the day when it is frozen. Particularly striking is the blue curve ("2020–05-01"): on the left, the activity participation level is frozen at 71% (cf. Table 3), while on the right it varies mostly between 80% and 100% as given by the data plotted in Fig 7. This is consistent with the theoretical argument (Sec. Decreasing marginal effect of interventions) that an activity participation of 71% reduces infections to $71\% \cdot 71\% \approx 50\%$ while an activity participation of 90% reduces infections only

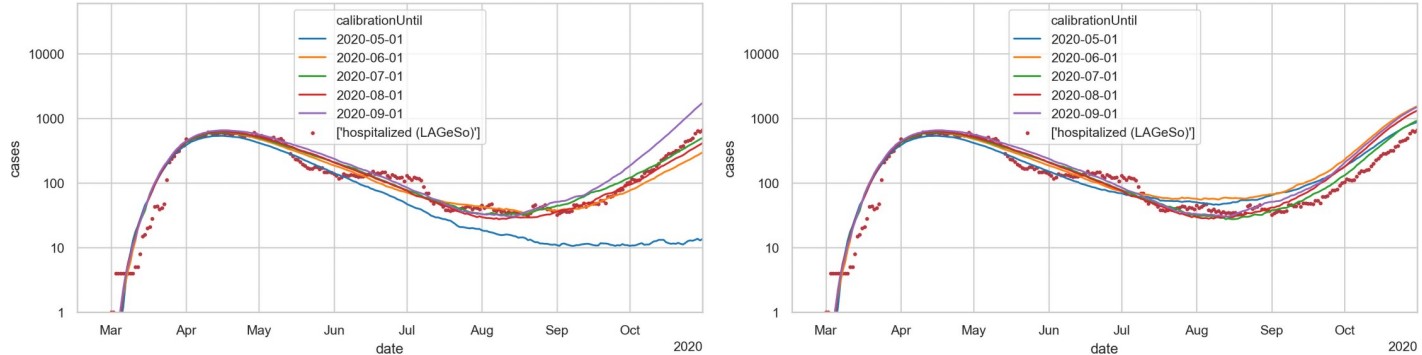

**Fig 17. Hospitalized persons for different calibration runs compared to real data.** Θ is calibrated such that hospital numbers in the simulation match the real data (red dots) until different points in time as indicated by the legend. After this date, an out of sample prediction is carried out. Until the calibration date real weather and disease import data is used. After the calibration date average weather data from the past ten years is used and the disease import is set to 4 imported cases per day (1 agent per day) LEFT: Activity levels are frozen at the level of the last day of the period used for calibration. RIGHT: Real activity levels are used.—Results are averaged over 30 Monte Carlo seeds.

to 91% · 90% ≈ 80%. That difference is sufficient to generate the difference between the two curves.

## Discussion

### Intuition for these results

In an older version of the model [33], we had all contact intensities set to one. The contributions of each activity type to the infection dynamics then in first order corresponded to the average weekly time consumption in the respective activity. For example, averaged over the week including the weekend, school consumes about 5 hours per day for persons going to school. However, since in Berlin only about 10% of the population are school children, https://www.statistik-berlin-brandenburg.de/BasisZeitreiheGrafik/Bas-Schulen.asp?Ptyp=300&Sageb=21001&creg=BBB&anzwer=5 the average time consumption for the school activity is only 0.43 hours per day when taken across the whole population (cf. Table 4). In contrast, there are more persons going to work than to school, thus increasing the weight of work in the infection dynamics (1.83 hours per day). A similar weight comes from the leisure activities, which are not necessarily more hours per week for each individual person, but where *all* persons contribute to this type of time consumption, resulting in an average of 1.67 hours per day. In consequence, restricting leisure activities had a large effect in that model.

**Table 3. Calibration parameter Θ and activity participation for the different out of sample predictions shown in Fig 17.**

| run | Θ | activity participation (if activity level frozen) | training error | prediction error (frozen activity levels) | prediction error (real activity levels) |
|---|---|---|---|---|---|
| 2020–05–01 | 1.20e-05 | 71% | 0.372 | 5.500 | 0.34 |
| 2020–06–01 | 1.27e-05 | 88% | 0.252 | 0.187 | 0.74 |
| 2020–07–01 | 1.29e-05 | 90% | 0.232 | 0.047 | 0.11 |
| 2020–08–01 | 1.30e-05 | 90% | 0.200 | 0.033 | 0.37 |
| 2020–09–01 | 1.32e-05 | 96% | 0.181 | 0.442 | 0.52 |

RMSLE (= Root Mean Square Logarithmic Error) for the calibration interval (training error) as well as for prediction period between 09–01 and 10–31 (prediction error). The Optuna package does not return confidence intervals for estimated parameters.

**Table 4. Average time consumption of out-of-home activities.**

| Activity | average time consumption [hr] | normalized contact intensity $ci'$ | $time \cdot ci'$ |
|---|---|---|---|
| day care | 0.22 | 11 | 2.42 |
| schools | 0.43 | 11 | 4.73 |
| university | 0.13 | 5.5 | 0.75 |
| work (incl. business) | 1.83 | 1.47 | 2.69 |
| shop | 0.38 | 0.88 | 0.33 |
| errands | 0.77 | 1.47 | 1.14 |
| leisure | 1.67 | 9.24 | 15.43 |
| home | 16.45 | 1 | 16.45 |

Averaged over a full week including Saturday and Sunday. The remaining time is spent travelling between activities. Contact intensities are taken from Table 1.

In the present model, the time consumptions are now multiplied by the normalized contact intensities in those activity types, cf. Table 4. In consequence, leisure, which already had a large share before, is now multiplied with a large contact intensity, and in consequence now gets even more weight. Work, despite occupying similar amounts of time, is weighted down because of the multiplication by a much smaller contact intensity. On the other end of the scale, public transport has, at full occupancy, a large contact intensity, but the times spent in public transport are considerably smaller than, say, at work. Also, persons in public transport are required to wear masks, while at work they are not.

A complicated case are schools and day care: They occupy large amounts of time, *and* have a large contact intensity, both somewhat similar to leisure. In consequence, the re-opening of day care in July and of the schools in August should have had strong consequences in the infection numbers but did not. We took that observation as confirmation that their larger-than-average contact intensity is compensated for by a smaller-than-average infectivity and susceptibility (cf. Sec. Children).

For other diseases, for example influenza, all of the above may need to be adapted. For example, children may have a larger infectivity/susceptibility than adults, which then multi-plied with their large contact intensity would lead to a large contribution to the infection dynamics. In consequence, these sub-models need to be understood and re-calibrated for each individual communicable disease.

## Robustness

The simulation uses one uniform $\Theta$ (cf. Eq 1) that remains the same over the whole simulation period. In consequence, the dynamics is driven by other inputs. These are, after the initial disease import (Fig 5), primarily the activity participation (Fig 7) and the temperature-dependent outdoors fraction (Fig 9).

The importance of the activity participation can be taken from Fig 17, where the blue curve ("2020–05–01") uses exactly the same setup on the left and on the right except for the activity participation level, which on the left is frozen at 71%, and on the right varies mostly between 80% and 100% as given by the data plotted in Fig 7. That was already discussed at the end of Sec. Out of sample prediction.

The importance of the temperature effect can best be taken from the calibration sequence: Fig 6 showing that a $\Theta$ smaller than 0.6 is not plausible; Fig 8 showing that reduced activity participation alone is not able to bring infections and resulting hospital levels down to the observed level during summer; Fig 10 showing that a symmetric indoors/outdoors model is

able to achieve that but misses the fall wave; and finally Fig 11 showing that the asymmetric indoors/outdoors model, with a significantly higher threshold temperature at the end of the summer, is able to also generate the fall wave. We have checked for other mechanisms driving the fall wave but obvious possibilities, such as the return-to-school or the summer disease import, both make the hospital numbers in fall start going up in the middle of August rather than at the beginning of September. S4 Text also shows that a less asymmetric indoors/outdoors model performs worse. Evidently, it is possible that in reality the virus seasonality is also caused by other aspects [84], and our model absorbs them into the indoors/outdoors model. Still, if one accepts the aerosol infection as major pathway, then the assumption that indoors vs. outdoors activities play a strong role is plausible.

The effect of the contact intensities was already discussed in Sec. Intuition for these results. From that discussion, it follows that the number of infections in an activity context depends, in first order, linearly on the contact intensity of that activity type. In consequence, if a contact intensity in Table 1 or Table 4 is, say, a factor of 2 too large, then the resulting infections (Fig 13 top), share of infections (Fig 13 bottom), and R-value per activity type (Fig 15) in first order get divided by two. In second order, the calibration parameter Θ would have to get increased to bring infections back to the previous level.

S4 Text also shows that the model fit degrades considerably when the mask model is removed.

## Comparison to other models

A comparison to compartmental models, in particular to the model of Chang et al. [10] which is at the border between compartmental and agent-based, can be found in S5 Text. The same text contains also a comparison with the model by Aleta et al. [21].

## Comparison to other "reductions of *R*" studies

Table 5 extracts "additional reductions to *R*" from other studies and compares them to our results. One immediately finds two issues: (A) The categories are not well aligned. For example, "small gathering cancellation" refers to gatherings with 50 persons or less, while other studies cancel gatherings *larger* than a certain number. Again other studies just consider a "gathering ban", but at the same time have "event ban" and "venue closure" as separate items. (B) Even where the categories are well aligned, the resulting numbers vary significantly: for example, "closing schools and universities" goes from 16% to 38%.

In part, this is a consequence of the fact that the interventions are not standardized: For example, the number of exemptions in what is called a lockdown varied quite a lot between countries.

Additionally, the transmission mechanisms from policy decision to execution vary, so even if the concept may be the same, the effect may be quite different between countries. For example, our reductions to *R* caused by school closures come out at the lower end of the range, and lower than those of Brauner et al. [5]. We attribute this to the following two elements: First, the model by Brauner et al. has no initial disease import which is then brought to a halt. In consequence, their approach has to assign all changes in the infection dynamics to the school closures. The school closures in Berlin, with Mar/12 (fri) or Mar/15 (mon) as the last day of school, too late to explain that the infection numbers stopped in the middle of March. Also, Dehning et al. [87] have an additional change point on Mar/7, corroborating that something has changed before the school closures. Second, other than both Brauner et al. and Dehning et al., we have the mobility data of Fig 7 at our disposal. It is clear that there was considerably more societal adaptation around the weekend of Mar/13–14 than just keeping children at

**Table 5. Percent reduction of *R* in other studies.**

| Measure | Brauner et al. (1st wave) [5] | Sharma et al. (2nd wave) [85] | Haug et al. "CC" [86] | Haug et al. "Lasso" [86] | Our model (abs.) (Table 2) | Our model R = 2.24 |
|---|---|---|---|---|---|---|
| Closing schools and universities | 38% | | 16% | 21% | 0.38 | 21% |
| Closing educational institutions (after implementing protective measures) | | 7% | | | | |
| Closing schools | | | | | 0.15 | 7% |
| Closures of businesses | | 35% | | | | |
| Closure of work sector | | | | | 0.17 | 8% |
| Closing some high-risk face-to-face businesses | 18% | | | | | |
| Closing most nonessential face-to-face businesses | 27% | | | | | |
| Closing retail and close contact services | | 12% | | | | |
| Closure of retail and errands sectors | | 12% | | | 0.09 | 4% |
| Gatherings limited to ≤ 1000 | 23% | | | | | |
| Mass gathering cancellation | | | 33% | 0% | | |
| Gatherings limited to ≤ 100 | 34% | | | | | |
| Gatherings limited to ≤ 10 | 42% | | | | | |
| Small gathering cancellation | | | 35% | 22% | | |
| Closures of gastronomy | | 12% | | | | |
| Closures of night clubs | | 12% | | | | |
| Leisure and entertainment venues | | 3% | | | | |
| Banning all leisure activities (including gastronomy and private visits) | | | | | 1.04 | 92% |
| Night time curfew | | 13% | | | | |
| Stricter mask policy (mandatory in most or all shared/public spaces) | | 12% | | | | |

Percentages are rounded to integers. To the right are our own results, first in absolute reductions of *R*, then in percent reductions of *R* applied to an *R* of 2.24 (the overall *R* in the model where these values were taken). Evidently, for a smaller *R*, our percentage values would be higher.

home. Brauner et al. themselves write that "the closure of schools . . . may have caused . . . behaviour changes. We do not distinguish this indirect signalling effect from the direct effect". Additionally, in Germany, children staying at home will force their parents to stay at home, thus forcing them into home office. In consequence, some of this may not be signalling, but causal secondary effects. In consequence, our model is more differentiated: What Brauner et al. attribute to the school closures alone is in our model attributed to a combination of school closures, behavioral changes, and the reduction of various other out-of-home activities. Thus, all of the values may be correct: The pure effect of school closures in western countries (with relatively few young people) may not be larger than 7%, but the measurable consequence for *R* when governments closed schools as their first intervention presumably indeed was much larger.

Clearly, data-driven mechanical models such as ours help clarifying the categories since we can exactly specify what we mean by closing some activity type or wearing a mask at certain activity types. Also, we can differentiate between the transmission from political decision to behavioral execution vs. the consequences of the behavioral execution to the infection dynamics. Finally, we can mechanically include organizational approaches such as contact tracing.

## Masks

We have checked our relatively large reductions of $R$ for masks in Table 2 multiple times. They are a consequence of the assumption that N95 masks reduce intake to 2.5%, taken from [67]. The review article [88] comes up with about 5%, a factor of two larger, but still displaying a very large reduction. The same paper [88] also shows that "masks" without a specification of the type has much less of an effect. Finally, there may be the issue that lay people may not be able to use N95 masks at full efficiency. In consequence, our results have to be interpreted once more "mechanically": They are plausible under the assumption that the fraction of people specified in the model is indeed able to use N95 masks effectively.

## Under-reporting, and its variation over time

A known issue with epidemiological data and thus the simulations that build on it is the issue of under-reporting, i.e. that there are more infections in reality than are in the data. Looking at Fig 12, it is clear that our current model assumes only little under-reporting during August to October. This originally led to hospital numbers that were too large; since we cannot reduce the number of infections below the case numbers, this justifies why we reduce simulated hospital numbers by a factor of 2 compared to [15] (cf. Fig 3). This, in turn, implies that, if we want to get the spring hospital numbers right, our simulated infection numbers in spring need to be about a factor of 8 larger than the reported case numbers.

Also note that our simulation includes non-symptomatic cases, which come on top of the symptomatic cases that we show in our figures such as Fig 12; that is, the actual under-reporting is even larger. Still, it is entirely possible that Germany's testing strategy is missing even more cases, in which case the simulation would need to aim for even larger numbers of infected persons. As long as the number of seropositive persons in Germany remains in the single-digit percentage ranges [89], the predictions made by the simulation are not strongly affected by this issue. Once the infections start to saturate, i.e. approach herd immunity, this will become important. Hopefully, by then systematic antibody screenings will be available, and we will be able to calibrate the model against the case numbers that must have been infected in the past. Given that we have the hospital numbers for control, we expect this to be straightforward.

## Making the model more realistic

Evidently, the model can be made (even) more realistic. Important aspects are the adaptation of the daily schedules to to restrictions, the dependence on income, and more realistic contact structures. All three aspects are discussed in S6 Text.

## Policy advice

The model was and is used for policy advice. Our regular reports to the government all have a DOI, for example [3] or [4]. Again, see https://depositonce.tu-berlin.de/simple-search?query=modus-covid.

## Conclusions

We combine a person-centric human mobility model with a mechanical model of infection and a person-centric disease progression model into an epidemiological simulation model. Different from other models, we take the movements of the persons, including the intervening activities where they can interact with other people, directly from data, which has already been available for transport planning before the pandemics. For privacy reasons, we rely on a

process that takes the original mobile phone data, extracts statistical properties, and then synthesizes movement trajectories from the statistical properties; one could use the original mobile phone trajectories directly if they were available. The model is used to replay the epidemics in Berlin. It is shown that the second wave in Berlin can be modelled well with an explicit temperature dependency of the outdoors fraction for leisure activities. The model is then used to evaluate different intervention strategies, such as closing educational facilities, reducing other out-of-home activities, wearing masks, or contact tracing, and to determine differentiated changes of the reproduction number *R* per intervention.

## Supporting information

**S1 Appendix. Appendix.**
(PDF)

**S1 Fig. Reduced activity participation by activity type.**
(PDF)

**S1 Text. Senozon method.**
(PDF)

**S2 Text. Model history.**
(PDF)

**S3 Text. Error metric for calibration.**
(PDF)

**S4 Text. Robustness runs.**
(PDF)

**S5 Text. Comparison to other models.**
(PDF)

**S6 Text. Making the model more realistic.**
(PDF)

**S1 File.**
(PDF)

## Acknowledgments

We thank Kai Martins-Turner, Dominik Ziemke, Tim Conrad and Natasa Conrad for frequent inputs and discussion. We are grateful to BVG (Berlin public transit operator) for providing the mask compliance rates which they surveyed on a daily basis. The work on the paper was funded by the Ministry of research and education (BMBF) Germany (01KX2022A) and TU Berlin; regular reports can be found through this search: https://depositonce.tu-berlin.de/simple-search?query=modus-covid. Zuse Institute Berlin (ZIB) provided CPU time.

## Author Contributions

**Conceptualization:** Sebastian A. Müller, Kai Nagel.

**Data curation:** Sebastian A. Müller, Michael Balmer, William Charlton, Ricardo Ewert, Andreas Neumann, Christian Rakow, Tilmann Schlenther.

**Formal analysis:** Sebastian A. Müller.

**Funding acquisition:** Kai Nagel.

**Investigation:** Sebastian A. Müller, Kai Nagel.

**Methodology:** Sebastian A. Müller, Kai Nagel.

**Project administration:** Ricardo Ewert.

**Software:** Sebastian A. Müller, Christian Rakow, Kai Nagel.

**Supervision:** Kai Nagel.

**Validation:** Sebastian A. Müller, Kai Nagel.

**Visualization:** Sebastian A. Müller, William Charlton.

**Writing – original draft:** Sebastian A. Müller, Kai Nagel.

**Writing – review & editing:** Sebastian A. Müller, Michael Balmer, William Charlton, Ricardo Ewert, Andreas Neumann, Christian Rakow, Tilmann Schlenther, Kai Nagel.

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
