## [Decision Letter · Decision Letter 0]

25 May 2021

PONE-D-21-11422

Predicting the effects of COVID-19 related interventions in urban settings by combining activity-based modelling, agent-based simulation, and mobile phone data

PLOS ONE

Dear Dr. Nagel,

Thank you for submitting your manuscript to PLOS ONE. After careful consideration, we feel that it has merit but does not fully meet PLOS ONE’s publication criteria as it currently stands. Therefore, we invite you to submit a revised version of the manuscript that addresses the points raised during the review process.

We look forward to receiving your revised manuscript.

Kind regards,

Itzhak Benenson, Ph.D.

Academic Editor

PLOS ONE

Journal Requirements:

'The work on the paper was funded by the Ministry of research and education (BMBF) Germany (01KX2022A) and TU Berlin.'

We note that one or more of the authors are employed by commercial companies: Senozon AZ and Senozon GmbH.

4. We note that Figure 1 in your submission contains copyrighted images.

All PLOS content is published under the Creative Commons Attribution License (CC BY 4.0), which means that the manuscript, images, and Supporting Information files will be freely available online, and any third party is permitted to access, download, copy, distribute, and use these materials in any way, even commercially, with proper attribution. For more information, see our copyright guidelines: http://journals.plos.org/plosone/s/licenses-and-copyright.

We require you to either (a) present written permission from the copyright holder to publish this figure specifically under the CC BY 4.0 license, or (b) remove the figure from your submission:

b. If you are unable to obtain permission from the original copyright holder to publish this figure under the CC BY 4.0 license or if the copyright holder’s requirements are incompatible with the CC BY 4.0 license, please either i) remove the figure or ii) supply a replacement figure that complies with the CC BY 4.0 license. Please check copyright information on all replacement figures and update the figure caption with source information. If applicable, please specify in the figure caption text when a figure is similar but not identical to the original image and is therefore for illustrative purposes only.

5. Please ensure that you refer to Figure 16 in your text as, if accepted, production will need this reference to link the reader to the figure.

Reviewers' comments:

Reviewer's Responses to Questions

**Comments to the Author**

1. Is the manuscript technically sound, and do the data support the conclusions?

Reviewer #1: Yes

Reviewer #2: Yes

2. Has the statistical analysis been performed appropriately and rigorously? 

Reviewer #1: Yes

Reviewer #2: Yes

3. Have the authors made all data underlying the findings in their manuscript fully available?

Reviewer #1: Yes

Reviewer #2: Yes

4. Is the manuscript presented in an intelligible fashion and written in standard English?

Reviewer #1: No

Reviewer #2: Yes

5. Review Comments to the Author

Reviewer #1: The paper “Predicting the effects of COVID-19 related interventions in urban settings by combining activity-based modeling, agent-based simulation, and mobile phone data” proposes a person-centric model that incorporates mobility patterns derived by cell phone usage and numerous other factors with a person-specific variant of a S(E)IR model to better understand how specific political interventions affect the rate of reinfections. Overall, the model offers an immaculate depth of a possible system that explains the progression of the pandemic in Berlin. Still, there are multiple major and minor questions that I would like to be answered before advising the manuscript for publication.

Major Remarks:

-While I generally admire the depth of the proposed model, I am not completely sure from what empirical evidence we really draw the conclusion. In particular, I am not sure to what extent the model choices or the mobility patterns drive the results. In my opinion, there are still multiple unknowns regarding the transmission of COVID-19, e.g., where do they happen, what effective difference is between an N95, medical mask, or cloth. Still, all this missing knowledge is completed with sometimes more or less empirical values in the presented model. I definitely see the sense of having this type of model, but the caveats should be stated clearly in the paper.

-The article would greatly benefit from a discussion of the sensitivity and stability of the system. If I understand the manuscript correctly, only one parameter is chosen in the Calibration, but how sensitive and stable is the system to different values of this coefficient? And what kind of sensitivity checks of all other parameters were carried out? By this I mean some assessment of how robust the conclusions drawn are.

An adequate discussion and communication of the uncertainties in the model are missing. On multiple occasions, this would benefit the manuscript.

- Fig. 4: What do the shaded areas indicate?

- Fig. 6 TOP: The numbers are really low, hence some uncertainty over multiple iterations of the pandemic would be great. Here, one could, e.g., sample different mobility patterns in the Senozon method? Overall, multiple runs of the simulations should enable some type of uncertainty quantification.

-The calibration of the only non-fixed parameter also has associated uncertainty. Here some additional information on how the calibration is performed is needed. Can one understand it as some type of method of moments, where the coefficient is picked so that the observed time series of cases and hospitalizations is conserved as closely as possible in the simulation model? If so it is unclear one differentiates between primary and secondary priority. Some mathematical formulas in the Annex or Supplementary Material would be helping.

In the article prediction using the model is named as a possible application. Therefore, some out-of-sample results using historical data are needed.

-What are the effects of the multiplicative form in equation (1)? Doesn't this form assume that all components are inherently on the same scale? Even though the approximation in (2) is only used for illustrative reasons when does the approximation exactly hold?

Minor Remarks:

-Generally, the presentation of the introduction could be more concise. At the moment, the narrative is based on how prior versions of the model were improved over time with multiple features but it does not become apparent why we even need a person-centric simulation model next to the standard S(E)IR models readily available. Also, I would check if all the citations to prior versions of the model are really needed.

It is also not clearly evident what the section on related work really wants to achieve. A clear motivation of agent-based models would be more than enough but the different types of mobility data should rather be discussed in the Annex.

-Senozon method: Some illustrative examples with a Figure akin to Fig. 1 would be a good addition to the explanation. Is for each person one mobility pattern sample for each day? If so how is prediction performed, where no observations of mobility patterns are available? Also, the section would be more meaning full if some toy examples of an origin-destination matrix and the raw data could be provided. What percentage of people is included in the raw data? Some caveats on the uncertainty associated with the plausible classifications of activities would be fitting. Finally, to be it is not clear how people assumed to be meeting one another given two specific meeting patterns.

-While the title of the manuscript suggests that the focus lies on urban areas, in the text, we find repeated comments that complete Germany is modeled as well. Why are only the results for Berlin reported? Does the sample only include urban areas or what data sources are missing for rural areas?

-Does the simulation assume that each home, school etc has the same constant values named in Table 1? Is this reasonable? Even in Berlin, there is some type of income gap, that probably leads to more space per person in wealthy neighborhoods. How could this affect the findings of the article? A random distribution could be used as an alternative maybe even based on the ZIP code?

-Are the lines in Fig. 4 in- or out-of-sample predictions of the model? And is the blu line the number of people with symptom onset? If so why is it so far of especially in the first few months? In addition, the temporal path of the R-value (Fig. 7) compared to the officially reported value by the RKI would be great to see and assess the proposed model.

-Did I understand the reasoning behind the additive strucure of interventions correctly in that it is based on the definition as a sum over individual reinfections? If so this should be stated more clearly in the manuscript.

-Table 3 is not clear to me. The authors of each study should be directly named together with some classification of the used method. Further, the proposed model should be one column in the table. Also, the star at 0 is not clear to me, how can 0 be significantly different from 0?

Reviewer #2: The manuscript “predicting the effects of covid-19 related interventions in urban settings by combining activity-based modeling, agent-based simulation, and mobile phone data” presents a novel approach to simulate the spread of epidemics in urban environments. As opposed to compartmental models, the model presented in the manuscript presents a personalized spatially explicit framework to track the contact between individuals, the probability of infection in each contact, and the progression of the disease in those individuals who were infected. The model manages to predict the trend of the disease spread in the Berlin metropolitan area. As such, I enjoyed reading the manuscript, as it presents a breakthrough in modeling epidemics, by allowing the insertion of explicit and detailed policy measures.

However, the manuscript contains pitfalls and is often unclear. The authors must take this into account in order for the manuscript to make a valid and clear scientific claim regarding the spread of epidemics in urban environments.

Model fit:

The calibration process of the model manages to fit the model with the general trend of the number of cases and hospitalized patients in berlin. However, the manuscript doesn’t suggest any metric that evaluates the model fit. Even if this kind of metric is not the purpose of the authors’ model evaluation, figure 4 presents the data in a logarithmic scale, in order to show the fit of general trend. However, this type of presentation is a bit deceiving without explicitly informing the reader that the purpose of the figure is to compare the trend and not absolute numbers.

Simulating agents activity:

It is clear that the COVID-19 pandemic changed the travel habits of nearly the entire population, in both short-term and long-term habits. It seems that the simulation conducted takes into account only the short-term travel habits, as it does not simulate replanting of agents' plans of mode choice and destination choice, such as changing the activities to telecommuting and teleducation. Choosing a different mode of transport is very likely given lower capacity or frequency in PT systems, and it is possible to rerun the simulation every n days in order to observe what happens to the modal split. It is true that simulating destination choice is very complex and might not be possible using this framework, however the authors should mention this limitation. In addition, it is not clear from the paper what happens when a person's activity is removed - how does the rest of its plan change?

Section - Infection model:

1. page 8: Equation 1 is not clear enough. M is the sum of which other persons? Those in contact? The explanation is for 2 individuals or for multiple individuals?

2. CI is different for each type of activity, due to the fact that people exhale and inhale more air in different activities (talking, eating, working out etc.). Was this taken into account? Though it affects the wedding rate(sh) more than the actual contact intensity, it is reasonable to materialize it through CI.

Page 9:

The explanation for room capacity was hard to grasp. If I understand correctly, nspaces reflects the fact that there is a smaller chance for two people to meet, however if they do, the chance of infection is higher.

Page 10:

1. Table 1 is unclear. What term does each column come to represent? The air exchange low and high calculation is not detailed in the main body of the text, and is unclear.

2. The calculation of CI needs to be refined in an orderly fashion. I tried to manually calculate the values in table 1 and failed.

3. What is the transition probability from being infectious to showing symptoms? It does not appear in the text.

Sub section - Indoors/outdoors and second wave:

Page 13:

Did leisure activities in Berlin in fact take place outdoors, or that they were also restricted? What fraction of the activities do leisure activities take up?

“Too early” “not steep enough” please refer to the specific location in the graph, by annotating the graph or by explicitly mentioning the coordinates on the graph.

Sub section - Infections per activity type:

Page 13:

The share of each activity frequency in both count and total duration across all agents is important in order to fully understand figure 6.

Sub section - Reductions of R per intervention:

Page 16:

Table 2 - is R calculated over the entire period? A certain date? If so please specify when.

When reporting reference [3], you used percent reduction, however here you use additive reduction while explaining why it is better - you should point out the weakness in percent reduction when first mentioning it(page 2).

Sub section - Intuition for these results:

Page 17:

The intuition for decreasing marginal effect of interventions should be presented in a formal way.

A graphical explanation of intuition for the results would improve the understanding of that part.

General remarks regarding the organization of the paper:

I don’t fully understand the motivation of the authors to move the calibration section to the appendix. It seems that the calibration is a fundamental part of the methods section. Moreover, the presence of the calibration sections in the appendix makes it difficult for the reader to understand the various abbreviations and references that appear later on.

Additionally, the section that discusses simulation runs should be a part of the methods section. The “Decreasing marginal effect of interventions” and “Intuition for these results” subsections should be a part of the discussion, as they reflect upon the results and do not report them.

6. PLOS authors have the option to publish the peer review history of their article (what does this mean?). If published, this will include your full peer review and any attached files.

Reviewer #1: No

Reviewer #2: No

---

## [Author Response · Author response to Decision Letter 0]

4 Jul 2021

Responses to reviewers are contained in a separate pdf document.

---

## [Decision Letter · Decision Letter 1]

10 Aug 2021

PONE-D-21-11422R1

Predicting the effects of COVID-19 related interventions in urban settings by combining activity-based modelling, agent-based simulation, and mobile phone data

PLOS ONE

Dear Dr. Nagel,

Thank you for submitting your manuscript to PLOS ONE. After careful consideration, we feel that it has merit but does not fully meet PLOS ONE’s publication criteria as it currently stands. Therefore, we invite you to submit a revised version of the manuscript that addresses the points raised during the review process.

Please react to the last remarks of the reviewer #1 trying to preserve the length of the paper or even making it shorter 

We look forward to receiving your revised manuscript.

Kind regards,

Itzhak Benenson, Ph.D.

Academic Editor

PLOS ONE

Journal Requirements:

Additional Editor Comments (if provided):

Please react to the rest of the Reviewer #1 comments preserving the length of the paper or even making it shorter

Reviewers' comments:

Reviewer's Responses to Questions

**Comments to the Author**

1. If the authors have adequately addressed your comments raised in a previous round of review and you feel that this manuscript is now acceptable for publication, you may indicate that here to bypass the “Comments to the Author” section, enter your conflict of interest statement in the “Confidential to Editor” section, and submit your "Accept" recommendation.

Reviewer #1: (No Response)

Reviewer #2: All comments have been addressed

2. Is the manuscript technically sound, and do the data support the conclusions?

Reviewer #1: Partly

Reviewer #2: Yes

3. Has the statistical analysis been performed appropriately and rigorously? 

Reviewer #1: No

Reviewer #2: Yes

4. Have the authors made all data underlying the findings in their manuscript fully available?

Reviewer #1: Yes

Reviewer #2: Yes

5. Is the manuscript presented in an intelligible fashion and written in standard English?

Reviewer #1: Yes

Reviewer #2: Yes

6. Review Comments to the Author

Reviewer #1: The revision addressed most of my raised points from my review. However, I still see some possible drawbacks of the proposed method, which I would like to be replied to before advising for its publication:

- I like the added section on Out-of-Sample prediction; however, some benchmark comparison to other models (SEIR or mechanistic models, like yesterday is today) would be interesting.

- Why was the RMSLE used for prediction? As far as I know, this is no standard measure. Why not use the RMSE or some proper scoring rule?

- Could you add some explanation or comments on two findings of the predictive assessment: Why is the prediction error in the first run under frozen activity levels so high? And it also seems as if updating the activity levels did, in fact, not lead to a better performance in all runs. What does this tell us about the model? Maybe some plots on how the activities changed in that time would be a starting point for the explanation?

- Generally, the size of the Figures should be consistent. Fig. 17 is a lot smaller than, e.g., Fig. 18

- While I admire the additional work put into extending the paper, I believe that 30 Pages is too long for the main article. I would put some parts of the discussion into the Appendix and most parts of the Appendix into an Online Supplementary Material (if this is possible with Plos One). Maybe also some of the cut discussion from the previous sections “Using mobile device data to observe changes of mobility behavior during COVID-19” and “From reductions of mobility behavior to reductions of infections” could be interesting in the Appendix of the Paper.

- For completeness, a short explanation of the Whitepaper of the Senozon Method in English (the language of the article itself) would be helpful. Besides, some rationale and legitimisation would benefit the article on why a model for transportation is also suitable for modelling COVID hospitalisation

Reviewer #2: I have read the authors reviewed manuscript and responses and find them satisfactory. Therefore, I recommend accepting the manuscript for publication.

7. PLOS authors have the option to publish the peer review history of their article (what does this mean?). If published, this will include your full peer review and any attached files.

Reviewer #1: No

Reviewer #2: No

---

## [Editor Report · Decision Letter 2]

12 Oct 2021

Predicting the effects of COVID-19 related interventions in urban settings by combining activity-based modelling, agent-based simulation, and mobile phone data

PONE-D-21-11422R2

Dear Dr. Nagel,

We’re pleased to inform you that your manuscript has been judged scientifically suitable for publication and will be formally accepted for publication once it meets all outstanding technical requirements.

Kind regards,

Itzhak Benenson, Ph.D.

Academic Editor

PLOS ONE
---

## [Editor Report · Acceptance letter]

20 Oct 2021

PONE-D-21-11422R2 

Predicting the effects of COVID-19 related interventions in urban settings by combining activity-based modelling, agent-based simulation, and mobile phone data 

Dear Dr. Nagel:

I'm pleased to inform you that your manuscript has been deemed suitable for publication in PLOS ONE. Congratulations! Your manuscript is now with our production department. 

Kind regards, 

on behalf of

Professor Itzhak Benenson 

Academic Editor

PLOS ONE